# QiMeng-CodeV-R1: Reasoning-Enhanced Verilog Generation

**Yaoyu Zhu**[1], **Di Huang**[1✉], **Hanqi Lyu**[1,2], **Xiaoyun Zhang**[1,3], **Chongxiao Li**[1,3],
**Wenxuan Shi**[1,3], **Yutong Wu**[1,3], **Jianan Mu**[1], **Jinghua Wang**[3], **Yang Zhao**[1,3],
**Pengwei Jin**[1], **Shuyao Cheng**[1], **Shengwen Liang**[1], **Xishan Zhang**[1,4],
**Rui Zhang**[1], **Zidong Du**[1], **Qi Guo**[1], **Xing Hu**[1], **Yunji Chen**[1,3]

[1] State Key Lab of Processors, Institute of Computing Technology, CAS
[2] University of Science and Technology of China
[3] University of Chinese Academy of Sciences
[4] Cambricon Technologies

https://iprc-dip.github.io/CodeV-R1

## Abstract

Large language models (LLMs) trained via reinforcement learning with verifiable reward (RLVR) have achieved breakthroughs on tasks with explicit, automatable verification, such as software programming and mathematical problems. Extending RLVR to electronic design automation (EDA), especially automatically generating hardware description languages (HDLs) like Verilog from natural-language (NL) specifications, however, poses three key challenges: the lack of automated and accurate verification environments, the scarcity of high-quality NL–code pairs, and the prohibitive computation cost of RLVR. To this end, we introduce CodeV-R1, an RLVR framework for training Verilog generation LLMs. First, we develop a rule-based testbench generator that performs robust equivalence checking against golden references. Second, we propose a round-trip data synthesis method that pairs open-source Verilog snippets with LLM-generated NL descriptions, verifies code–NL–code consistency via the generated testbench, and filters out inequivalent examples to yield a high-quality dataset. Third, we employ a two-stage "distill-then-RL" training pipeline: distillation for the cold start of reasoning abilities, followed by adaptive DAPO, our novel RLVR algorithm that can reduce training cost by adaptively adjusting sampling rate. The resulting model, CodeV-R1-7B, achieves 68.6 % and 72.9 % pass@1 on VerilogEval v2 and RTLLM v1.1, respectively, surpassing prior state-of-the-art by 12∼20 %, while even exceeding the performance of 671B DeepSeek-R1 on RTLLM. We have released our model, training code, and dataset to facilitate research in EDA and LLM communities. [1]

## 1 Introduction

Large language models (LLMs) have recently demonstrated remarkable progress on reasoning tasks when trained via reinforcement learning with verifiable reward (RLVR). Notable examples include OpenAI-o1 [26] and DeepSeek-R1 [3], which exhibit emergent reasoning capabilities on problems endowed with explicit verification procedures—such as software programming and mathematical problem solving. This success suggests a promising opportunity to apply RLVR for electronic design

---

✉ Corresponding author. Contact: {zhuyaoyu, huangdi, huxing}@ict.ac.cn.
[1]Please refer to https://iprc-dip.github.io/CodeV-R1/ for relative resources.

automation (EDA), specifically to the automatic generation of hardware description languages (HDLs) like Verilog from natural-language (NL) specifications [38].

However, the three foundational components required for effective RLVR — (i) a reliable verification environment, (ii) high-quality NL-code data, and (iii) an efficient training algorithm — each present significant challenges in training reasoning LLMs for Verilog generation:

**(1) Automated verification of hardware designs remains difficult.** RLVR requires a verification environment capable of providing accurate rewards. However, even in the data-rich software coding domain, such environments are rare. For example, most problems in the programming-contest dataset APPS [6] have only one or two sets of unit tests, and they exhibit a false-positive rate of up to 60% when evaluated with an average of 20 unit tests [12]. Consequently, the software community has adopted the practice of using LLMs to generate additional unit tests in order to improve verification quality [11, 41]. Nevertheless, this approach is both costly and of limited effectiveness for hardware designs, because LLMs lack the hardware-specific knowledge needed to handle the complex state spaces and corner cases of sequential circuits. For example, if the reset and clock signals are not correctly configured, the intended functionality cannot be properly verified.

**(2) High-quality NL–code pairs for hardware designs are scarce.** The proprietary nature of hardware designs severely limits the availability of annotated Verilog examples. Although several LLM-based methods have been proposed to synthesize NL–code pairs [5, 16, 45, 48], the resulting datasets often suffer from low-quality data (see Appendix D for examples), rendering them inadequate for RLVR's stringent requirements.

**(3) The computational cost of RLVR is prohibitive.** Training a 32B LLM on 1K data for 5 epochs using 16 NVIDIA H100 GPUs with supervised fine-tuning (SFT) takes only 0.5 hours [24]. In contrast, training a 14B LLM on 24K verifiable coding problems with reinforcement learning can take over 2.5 weeks on 32 NVIDIA H100 GPUs [20], making it prohibitively expensive to train a Verilog reasoning LLM using RLVR.

To overcome these challenges, we introduce CodeV-R1, a comprehensive RLVR framework for Verilog generation. Our contributions are threefold:

**(1) Automated testbench generation.** We develop a rule-based testbench generation framework to verify the equivalence between a given Verilog implementation and its golden reference as accurately as possible. For each golden reference, the framework first performs circuit-structure analysis to extract information such as input/output (I/O) ports and reset/clock signals. It then enumerates all reset and clock-synchronization scenarios to improve verification accuracy. Experiments demonstrate that our testbench achieves 96.1 % fewer false negatives than the LLM-generated counterpart and detects 62.5 % more injected errors in fuzzing tests for sequential circuits. Detailed experimental results are presented in Section 3.3.4.

**(2) Round-trip data synthesis for high-quality NL–code pairs.** Leveraging our testbench generation framework, we propose the round-trip data synthesis approach that can automatically synthesize high-quality NL–code pairs from code snippets. Specifically, candidate code snippets are first paired with LLM-generated NL descriptions, and then verified by regenerating the code from NL and comparing against the original for equivalence with our testbench. Only code that passes the testbench is retained and combined with the NL to form high-quality data for reinforcement learning. We theoretically prove that, given strong LLMs and an ideal verification environment, this procedure yields NL–code pairs of sufficiently high quality for RLVR with a high probability.

**(3) Two-stage training with adaptive DAPO for cost-effective RLVR.** We adopt a two-stage "distill-then-RL" training pipeline to cold-start LLMs' reasoning ability through SFT and apply RL to enhance model's reasoning ability. Specifically, we use DeepSeek-R1 as the NL-to-code LLM in our round-trip data synthesis to produce (NL, Thought, Code) triplets, based on which we perform SFT on our base LLM to obtain a distilled LLM with basic reasoning ability. Then, we apply RLVR on the distilled LLM using the equivalence-checked high-quality data to further enhance its Verilog generation capability. Additionally, recognizing that RLVR's bottleneck lies in sampling and verification [20], we extend dynamic sampling policy optimization (DAPO) [44] with an adaptive mechanism that dynamically adjusts the number of samples per training step based on past sample discard rates. This approach notably reduces unnecessary sampling and verification overhead, thereby achieving a 1.25x acceleration.

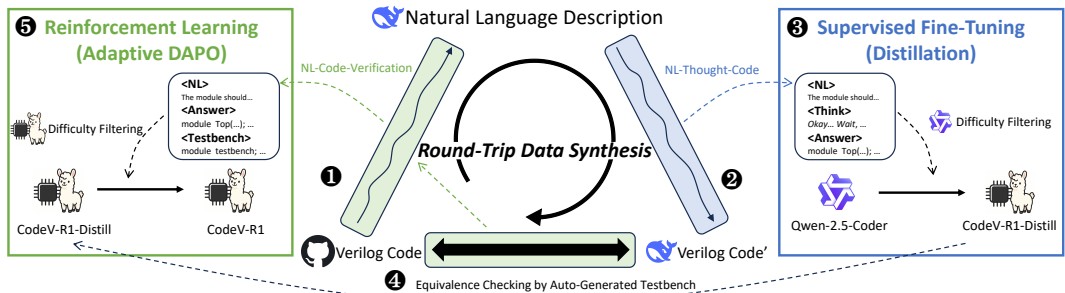

Figure 1: **The overview of CodeV-R1.** The core components of our framework include an automated testbench (Section 2.1), a supervised fine-tuning process (Section 2.2), and a reinforcement learning process (Section 2.3).

Based on these techniques, we develop CodeV-R1-7B, a specialized reasoning LLM for Verilog generation with only around 2,656 A100-GPU-hours. On the VerilogEval v2 [28] and RTLLM v1.1 / v2 [19] benchmarks, CodeV-R1-7B achieves 68.8% / 72.9% / 68.0% pass@1, respectively. Remarkably, it surpasses the 671B DeepSeek-R1 by 8.1% on RTLLM v1.1 and 3.3% on RTLLM v2, demonstrating its strong RTL generation capabilities.

## 2 Methods

Our framework comprises 5 stages, including an automated testbench generation framework (Figure 1). Stages ❶ $\sim$ ❸ constitute the distillation phase, and stages ❹ and ❺ comprise the reinforcement learning phase. Below, we introduce the processes of these phases:

❶ **Code-to-NL.** Following prior work [47, 48], we collect Verilog code snippets from GitHub (denoted $y^*$) and use an LLM (DeepSeek-V3 [4]) to produce corresponding natural-language summaries (denoted $x$), creating an NL–code corpus $\{(x_i, y_i^*)\}$ with approximately 150K data samples.

❷ **NL-to-Code.** Using DeepSeek-R1, we take each NL description $x_i$ from stage 1 and generate the "thought" (denoted $c_i'$) as well as an Verilog code snippet (denoted $y_i'$), producing NL–thought–code triples $\{(x_i, c_i', y_i')\}$.

❸ **Supervised Fine-Tuning.** We first filter the $\{(x_i, c_i', y_i')\}$ dataset by removing any examples for which base LLMs (*e.g.*, Qwen2.5-Coder-7B-Instruct / Qwen2.5-Coder-32B-Instruct [9]) can generate correct code in any of 5 attempts (correctness is verified using our automatically generated testbench). We then perform SFT using these data on the base LLM to bootstrap their reasoning ability, yielding the distilled model, CodeV-R1-7B-Distill. This stage uses approximately 87K examples.

❹ **Equivalence Checking.** We use our automated testbench to verify equivalence between the original snippets $y^*$ and the newly generated snippets $y'$. Any non-equivalent pairs $\{(x_i, y_i^*)\}$ are discarded, while equivalent pairs are retained as high-quality data for subsequent RL training.

❺ **Reinforcement Learning.** We again filter the retained $\{(x_i, y_i^*)\}$ set by removing any examples where the distilled model CodeV-R1-7B-Distill generates correct code in all of 5 attempts (as checked by the testbench). After this filtering, approximately 3.1K examples remain. We then apply our adaptive DAPO algorithm, a novel RLVR algorithm, to further improve Verilog-generation performance, resulting in the final model, CodeV-R1-7B. Next, we will describe in detail the automated testbench generation framework as well as the two training phases, distillation and RL.

### 2.1 Automated Testbench Generation Framework for Verilog Code

To facilitate the rule-based reward mechanism for the RL process, we have developed a specialized framework. This framework verifies the functionality of the generated Verilog code by conducting edge-triggered simulation and comparing it against the reference code. The verification framework unfolds in three consecutive phases:

**Phase 1: Circuit-Structure Analysis.** Before performing functional verification, we extract the input/output (I/O) ports along with their respective bit-widths from the reference golden code using

Yosys [39]. For sequential circuits, we identify clock signals, noting their edge polarity (rising or falling), and characterize reset signals through control flow analysis. Reset signals are categorized based on synchrony (synchronous if they depend on the clock) and polarity (active-high or active-low).

**Phase 2: Simulation.** We simulate by providing random inputs to both the generated and the reference codes, and evaluating the equivalence of outputs. For **combinational circuits**, we employ $M = 100$ independent simulation sequences for equivalence evaluation, each comprising $N = 1000$ inputs. Regarding **sequential circuits**, we adopt a dual-stage validation approach when dealing with circuits that have either one reset signal or no reset signal at all: Firstly, we execute simulations using $M = 100$ sequences, each with $N = 1000$ clock toggles (500 cycles) with randomized inputs. In this stage, deterministic reset signals—derived from golden reset behavior extracted via Yosys and representing expected, consistent reset logic—are applied at the start of each sequence, primarily aimed at testing the circuit's core functionality. Secondly, we conduct simulations with an identical number of sequences and clock cycles with random reset signals, which validates the consistency of the reset signal operation. For circuit designs featuring multiple reset signals, we exhaustively test every non-conflicting combination, all maintaining the aforementioned $\frac{MN}{2}$ cycle count.

**Phase 3: Verification.** After each clock toggle, we assess the equivalence of the outputs between the generated Verilog code and the reference implementation. This process results in a total of $2MN$ assessments for typical sequential circuits and $MN$ assessments for combinational circuits. The verification outcome is quantified by an error rate metric $\epsilon = \frac{\text{Error Number}}{2MN} \times 100\%$. A value of $\epsilon = 0\%$ indicates that the generated code functions correctly within our testbench environment. Through 32-way parallelization, the simulation achieves a throughput of 15 instances per second.

## 2.2 CodeV-R1-7B-Distill: Supervised Distillation for Verilog Data

Our pipeline for distillation begins with a set of Verilog code (denoted as $y_i^*$) collected from GitHub. We use DeepSeek-V3 to summarize these code snippets, producing instructions $x_i$ corresponding to $y_i^*$ (stage ❶). Then, to produce the corpus for distillation, we ask DeepSeek-R1 to generate responses containing "thought" $c_i'$ and Verilog code snippet $y_i'$ (stage ❷). These two stages yield approximately 150K NL-thought-code triples $(x_i, c_i', y_i')$.

Next, we curate a *challenging subset* through two filters: (1) retaining only instructions where baseline models (Qwen2.5-Coder-7B-Instruct / Qwen2.5-Coder-32B-Instruct) fail to generate the code passing the functional verification (Section 2.1) to $y_i^*$, and (2) ensuring synthesizability of $y_i^*$ with Yosys [39]. In addition, to prevent benchmark contamination, we remove samples where the generated code $y_i'$ exhibits Rouge-L similarity > 0.5 [13] to VerilogEval v1 [14] / v2 [28] or RTLLM v1.1 [19] / v2 [17], yielding 87K high-quality samples (stage ❸).

Finally, we initialize CodeV-R1-7B-Distill from Qwen2.5-Coder-7B-Instruct and fine-tune it to generate complete responses $(c_i', y_i')$ given $x_i$. Following DeepSeek-R1's methodology [3], we maximize the likelihood of the generated responses using our prompt template (see Appendix E), with implementation specifics detailed in Section 3.1 (stage ❸).

## 2.3 CodeV-R1-7B: Reinforcement Learning on the Distilled Model

To further improve the model's reasoning ability, we perform reinforcement learning fine-tuning based on CodeV-R1-7B-Distill with carefully selected high-quality Verilog data (stage ❹ and stage ❺). Below we will introduce our data curation method (Section 2.3.1), RL training algorithm, and reward design for RL (Section 2.3.2).

### 2.3.1 High-quality Data Curation

Experiences from prior research suggest that conducting RL training on problems that *the model can solve but requires reasoning to address* can more effectively enhance the model's RL capabilities [34]. Furthermore, given potential inconsistencies between the golden code $\{y_i^*\}$ in the original dataset collected from GitHub and the instructions $\{x_i\}$ generated by DeepSeek-V3, we prioritize *ensuring the correctness* of selected problems. To summarize, our RL (question, answer) pairs must meet three key criteria: being **solvable**, **challenging**, and **error-free**.

To implement this framework, we identify problems where DeepSeek-R1 *successfully generates code matching the golden one in the original dataset*, while both Qwen2.5-Coder-7B-Instruct and Qwen2.5-

Coder-32B-Instruct *fail to produce equivalent solutions*. Specifically, we conduct equivalence checking between the $\{y_i'\}$ code generated by DeepSeek-R1 in the 87K dataset and $\{y_i^*\}$ in the original dataset, retaining only *validated* $\{(x_i', y_i^*)\}$ pairs for RL training.

For difficulty enhancement, we employ CodeV-R1-7B-Distill to generate five code variants per question, excluding cases where all generated codes match the golden one, as these reflect patterns already mastered during supervised fine-tuning (stage ❹). Through this rigorous selection process, we curate a final dataset of 3.1K *high-quality* examples for reinforcement learning.

Additionally, we formalize the equivalence between code and natural language, and theoretically prove the effectiveness of our data curation. Intuitively, the Code-to-NL and NL-to-Code conversion process using LLMs inevitably leads to some information loss. Therefore, if the converted code remains equivalent to the original one after back-and-forth conversion, the probability of error during the conversion process is minimal. Detailed definition and proof are shown below.

**Definition 2.1** (NL-Code Deterministic Equivalence (NLCDE)). *Let $\mathcal{F}$ denote the space of all code snippets, $\mathcal{L}$ the space of natural-language (NL) descriptions, and $\mathcal{R} \subseteq \mathcal{F} \times \mathcal{L}$ a semantic/functional equivalence relation where $(f, l) \in \mathcal{R}$ iff code $f$ fully implements NL $l$ (or $l$ precisely describes $f$).*

*Consider two probabilistic models, $M_1 : \mathcal{F} \to \mathcal{L}$ (code-to-NL) and $M_2 : \mathcal{L} \to \mathcal{F}$ (NL-to-code), the* **NLCDE** *states: For all $f \in \mathcal{F}, l \in \mathcal{L}$: 1. If $M_1$ generates $l$ with $\Pr(l \mid f) = 1$, then $(f, l) \in \mathcal{R}$ (deterministic NL summaries are semantically equivalent to input code). 2. If $M_2$ generates $f$ with $\Pr(f \mid l) = 1$, then $(f, l) \in \mathcal{R}$ (deterministic code outputs are functionally equivalent to input NL).*

**Theorem 2.1** (Semantic Equivalence in Round-Trip Transformations). *Consider the probabilistic models $M_1 : \mathcal{F} \to \mathcal{L}$ (code-to-NL) and $M_2 : \mathcal{L} \to \mathcal{F}$ (NL-to-code) from the* **NL-Code Deterministic Equivalence (NLCDE)** *definition (Definition 2.1). Let $Y \in \mathcal{F}$ be a random code snippet drawn from some distribution, and define the transformed objects: $X = M_1(Y) \in \mathcal{L}$, $Y' = M_2(X) \in \mathcal{F}$. For any pair of objects $A, B$, let $E_{AB}$ denote the event "A and B are semantically equivalent." If the round-trip transformation preserves equivalence with certainty under NLCDE, i.e., $\Pr\big[E_{Y,Y'}\big] = 1$, then both forward and backward transformations are individually equivalent with certainty: $\Pr\big[E_{Y,X} \wedge E_{X,Y'}\big] = 1$.*

*Proof Sketch.* This theorem can be proved by the Data Processing Inequality. Please refer to Appendix A.1 for the detailed proof and further explanation. □

### 2.3.2 Adaptive DAPO Algorithm

We enhance the DAPO algorithm [44] with two efficiency improvements for RL fine-tuning on the distilled model (stage ❺). The core DAPO loss operates on groups of $G$ responses per prompt:

$$
\mathcal{L}_{DAPO}(\theta) = \mathbb{E}_{(x,y^*)\sim\mathcal{D}, \{y_i\}_{i=1}^{G}\sim\pi_{\theta_{old}}(\cdot|x)}
$$

$$
\left[ \frac{1}{\sum_{i=1}^{G}|y_i|} \sum_{i=1}^{G} \sum_{t=1}^{|y_i|} \min\left( r_{i,t}(\theta)\hat{A}_{i,t}, \mathrm{clip}\left(r_{i,t}(\theta), 1 - \epsilon_{low}, 1 + \epsilon_{high}\right)\hat{A}_{i,t} \right) \right],
$$

$$
\text{(1)}
$$

$$
\text{s.t.} \quad 0 < |\{y_i | \mathrm{is\_equivalent}(y_i, y^*)\}| < G,
$$

where $r_{i,t}(\theta) = \frac{\pi_\theta(y_{i,t}|x, y_{i,<t})}{\pi_{\theta_{old}}(y_{i,t}|x, y_{i,<t})}$, $\hat{A}_{i,t} = \frac{R_i - \mathrm{mean}(\{R_i\}_{i=1}^{G})}{std(\{R_i\}_{i=1}^{G})}$ ($R_i$ is the reward to be introduced later) , $r_{i,t}$ is the importance sampling ratio under the new policy $\pi_\theta$ compared to the old policy $\pi_{\theta_{old}}$, $\hat{A}_{i,t}$ is the group-relative advantage, $|y_i|$ is the length of response to calculate token-level loss, and $\epsilon_{low} < \epsilon_{high}$ are asymmetric clipping thresholds introduced in DAPO to encourage exploration. The constraint $0 < |\{y_i | \mathrm{is\_equivalent}(y_i, y^*)\}| < G$ ensures each training batch contains both correct and incorrect responses. Note that we do not include the overlong filtering proposed by DAPO here.

A key feature of DAPO is the dynamic sampling mechanism. It notably improves the training result. However, the standard DAPO sampling strategy presents inefficiencies during sample generation: DAPO's fixed generation batch size (denoted as $b_{gen}$) is suboptimal. If too few partially correct samples are generated for the RL train batch size (denoted as $b_{train}$), costly re-sampling occurs; if

too many are generated, excess samples are wasted. This problem intensifies as training progresses, and the improvement in model accuracy reduces the number of partially correct examples.

We address this with an **adaptive batch size** mechanism utilizing a dynamically estimated sampling effective ratio, $r_{valid}$. Initially, $b_{gen}$ is set to $b_{train}$. After successfully accumulating a full training batch ($b_{train}$), we calculate the batch effective ratio ($\frac{\text{number of valid samples}}{b_{gen}}$). The value of $r_{valid}$ is then updated to the *minimum* of itself and the batch effective ratio. For the subsequent sampling phase, the generation batch size is adaptively set to $b_{gen} = \lceil \frac{b_{train}}{r_{valid}} \rceil$. The detailed process is given in Appendix A.2. Note that this acceleration does not involve offline updates or alter the composition of the RL training batch, so it preserves DAPO's accuracy while accelerating training.

We implement a rule-based reward function that evaluates both structural correctness and semantic equivalence. A response $y_i$ receives a reward of 1 if it satisfies two conditions: (1) Proper formatting as "<think>reasoning</think><answer>solution</answer>" (2) Semantic equivalence with the golden code $y^*$ judged by the equivalence checker introduced in Section 2.1. The reward function $R(y, y^*)$ is 1 if $y$ has a correct format and $(y, y^*)$ are functional equivalent, and 0 otherwise.

## 3 Experiments

This section details the implementation of our method and presents comprehensive experimental results. We systematically evaluate our model through multiple dimensions: comparisons with prior state-of-the-art approaches, test-time scaling analysis across varying response length constraints, ablation studies analyzing the impact of golden code correctness and problem complexity, acceleration effects of the adaptive DAPO mechanism, and testbench performance evaluation. These analyses collectively demonstrate the effectiveness and efficiency of our proposed approach.

### 3.1 Implementation details

We obtain our final model by first distilling DeepSeek-R1 and then applying RL on our curated 3.1K dataset. During distillation, we employ LLaMAFactory [50] to supervised fine-tune (SFT) Qwen2.5-Coder-7B-Instruct using the 87K dataset filtered for distillation. We train the model for 6 epochs with a learning rate of $1 \times 10^{-5}$ and a batch size of 64. The total context length is set to 16384 during distillation. During RL, we use the verl [32] framework to further train the distilled model with our adaptive DAPO. We use a batch size of 128, a learning rate of $1 \times 10^{-6}$, and train for 300 steps. The rollout temperature is set to 1.0. During this stage, the max length is set to 2048 for instruction and 16384 for response. The SFT stage is executed on 8 A100-80G GPUs, taking approximately 78 hours, while the RL stage runs on 16 A100-80G GPUs, requiring around 127 hours of computation. The whole parameter setting is provided in Appendix B.

We test our distillation and RL model on various Verilog benchmarks, including VerilogEval v1 [14] / v2 [28] and RTLLM v1.1 [19] / v2 [17]. For VerilogEval v2, we examine zero-shot scenarios in both specification-to-RTL translation and code completion tasks. The maximum context length is configured to 16384 tokens during the evaluation phase for all benchmarks. The temperature during generation is 0.6 for the distillation model and 1.0 for the RL model, and 20 responses are generated per query to estimate the pass@k score for both VerilogEval and RTLLM.

### 3.2 Main Results

Our main experimental results are shown in Table 1 and Table 2. We evaluate DeepSeek-R1 [3], DeepSeek-V3 [4], QWQ-32B [36], DeepSeek-R1-Distill-Qwen-32B [3], DeepSeek-R1-Distill-Qwen-7B [3], Qwen2.5-Coder-32B-Instruct [42], Qwen2.5-Coder-7B-Instruct [42], and GPT-4o [25] on VerilogEval and RTLLM. Meanwhile, we adopt results reported by RTLCoder [16], BetterV [45], CodeV [48], CraftRTL [15] from their papers. The results demonstrate that:

**Our model achieves state-of-the-art (SOTA) performance among Verilog-domain models on most benchmarks.** Our model has a significant advantage over previous Verilog-domain models on RTLLM v1.1, outperforming the previous SOTA model, CraftRTL-DS-6.7B, by 18.8% on the pass@1 metric. On VerilogEval v1-Human, although the performance improvement compared to the previous SOTA model, CraftRTL-SC2-15B, is not substantial, our model has a smaller size (7B) compared

Table 1: Comparison of CodeV-R1-7B against baselines on VerilogEval v1 and RTLLM v1.1.

| Type | Model | Open source | VerilogEval-Machine (%) | | | VerilogEval-Human (%) | | | RTLLM v1.1 (%) | |
|---|---|---|---|---|---|---|---|---|---|---|
| | | | pass@1 | pass@5 | pass@10 | pass@1 | pass@5 | pass@10 | pass@1 | pass@5 |
| Foundation Models | GPT-4o* | × | 67.7 | 75.5 | 77.2 | 60.1 | 71.4 | 74.5 | 41.7 | 65.9 |
| | DeepSeek-R1-671B* | ✓ | 81.0 | 87.4 | 89.5 | 81.5 | 87.6 | 88.5 | 64.8 | 82.9 |
| | DeepSeek-V3-671B* | ✓ | 80.8 | 87.5 | 88.8 | 68.7 | 79.7 | 82.1 | 60.9 | 74.2 |
| | QWQ-32B* | ✓ | 71.1 | 84.0 | 87.0 | 63.8 | 78.0 | 81.3 | 50.9 | 70.6 |
| | DeepSeek-R1-Distill-Qwen-32B* | ✓ | 64.7 | 80.5 | 83.6 | 51.3 | 68.1 | 72.2 | 42.1 | 64.3 |
| | DeepSeek-R1-Distill-Qwen-7B* | ✓ | 5.3 | 16.9 | 24.9 | 1.6 | 6.3 | 10.1 | 0.0 | 0.0 |
| | Qwen2.5-Coder-32B-Instruct* | ✓ | 66.6 | 76.6 | 79.7 | 47.6 | 58.1 | 61.8 | 47.9 | 67.7 |
| | Qwen2.5-Coder-7B-Instruct* | ✓ | 60.2 | 77.8 | 82.4 | 31.9 | 46.3 | 50.2 | 32.2 | 48.2 |
| Specialized Models | RTLCoder-Mistral-7B | ✓ | 62.5 | 72.2 | 76.6 | 36.7 | 45.5 | 49.2 | - | 48.3 |
| | RTLCoder-DS-6.7B | ✓ | 61.2 | 76.5 | 81.8 | 41.6 | 50.1 | 53.4 | - | 48.3 |
| | BetterV-CL-7B | × | 64.2 | 75.4 | 79.1 | 40.9 | 50.0 | 53.3 | - | - |
| | BetterV-DS-6.7B | × | 67.8 | 79.1 | 84.0 | 45.9 | 53.3 | 57.6 | - | - |
| | BetterV-CQ-7B | × | 68.1 | 79.4 | 84.5 | 46.1 | 53.7 | 58.2 | - | - |
| | CodeV-CL-7B | ✓ | 78.1 | 86.0 | 88.5 | 45.2 | 59.5 | 63.8 | 39.4 | 62.1 |
| | CodeV-DS-6.7B | ✓ | 77.9 | 88.6 | 90.7 | 52.7 | 62.5 | 67.3 | 42.4 | 55.2 |
| | CodeV-CQ-7B | ✓ | 77.6 | 88.2 | 90.7 | 53.2 | 65.1 | 68.5 | 36.6 | 55.2 |
| | CraftRTL-CL-7B | × | 78.1 | 85.5 | 87.8 | 63.1 | 67.8 | 69.7 | 42.6 | 52.9 |
| | CraftRTL-DS-6.7B | × | 77.8 | 85.5 | 88.1 | 65.4 | 70.0 | 72.1 | 53.1 | 58.8 |
| | CraftRTL-SC2-15B | × | 81.9 | 86.9 | 88.1 | 68.0 | 72.4 | 74.6 | 49.0 | 65.8 |
| Ours | CodeV-R1-7B-Distill | ✓ | 76.2 | 85.6 | 87.0 | 65.7 | 76.8 | 79.7 | 57.4 | 75.8 |
| | CodeV-R1-7B | ✓ | 76.5 | 84.1 | 85.7 | 69.9 | 79.3 | 81.7 | 72.9 | 86.1 |

* We evaluate the models with *, while other results are sourced from their papers.

Table 2: Comparison of CodeV-R1-7B on VerilogEval v2 and RTLLM v2.

| Type | Model | Open source | VerilogEval2-SR (%) | | | VerilogEval2-CC (%) | | | RTLLM v2 (%) | | |
|---|---|---|---|---|---|---|---|---|---|---|---|
| | | | pass@1 | pass@5 | pass@10 | pass@1 | pass@5 | pass@10 | pass@1 | pass@5 | pass@10 |
| Foundation Models | GPT-4o | × | 64.1 | 73.7 | 76.2 | 57.6 | 66.1 | 69.0 | 56.5 | 70.3 | 75.2 |
| | DeepSeek-R1-671B | ✓ | 77.5 | 84.7 | 87.4 | 79.1 | 85.1 | 87.1 | 64.7 | 75.8 | 79.7 |
| | DeepSeek-V3-671B | ✓ | 62.4 | 71.7 | 75.0 | 68.7 | 76.3 | 78.2 | 59.1 | 71.5 | 73.3 |
| | QWQ-32B | ✓ | 64.2 | 77.3 | 80.1 | 64.0 | 77.8 | 80.9 | 52.9 | 68.0 | 71.2 |
| | DeepSeek-R1-Distill-Qwen-32B | ✓ | 43.9 | 63.3 | 69.2 | 53.8 | 69.8 | 73.8 | 42.4 | 62.1 | 67.0 |
| | DeepSeek-R1-Distill-Qwen-7B | ✓ | 0.6 | 2.2 | 3.5 | 2.0 | 7.0 | 11.3 | 0.0 | 0.0 | 0.0 |
| | Qwen2.5-Coder-32B-Instruct | ✓ | 47.5 | 60.7 | 64.7 | 46.6 | 59.0 | 62.8 | 47.8 | 63.9 | 67.8 |
| | Qwen2.5-Coder-7B-Instruct | ✓ | 31.3 | 49.3 | 54.6 | 30.5 | 46.8 | 52.0 | 36.1 | 52.4 | 57.6 |
| Specialized Models | RTLCoder-DS-6.7B | ✓ | 31.1 | 47.8 | 52.3 | 33.7 | 45.9 | 49.8 | 33.6 | 45.3 | 49.2 |
| Ours | CodeV-R1-7B-Distill | ✓ | 65.2 | 75.2 | 77.5 | 65.5 | 75.6 | 78.2 | 57.2 | 71.9 | 77.1 |
| | CodeV-R1-7B | ✓ | 68.8 | 78.2 | 81.1 | 69.9 | 78.2 | 80.9 | 68.0 | 78.2 | 81.7 |

* We evaluate all models in this table. SR: Specification-to-RTL; CC: Code Completion.

to theirs (15B). Among 7B models, we outperform the previous best model (CraftRTL-DS-6.7B) by 4.5% on pass@1. Although our model does not perform well on VerilogEval-Machine, this benchmark is relatively easy, and even DeepSeek-R1 does not have a significant advantage on it.

**Our model demonstrates superior performance over most foundation models across both benchmarks.** Although it does not surpass the DeepSeek-R1 model—the primary source for knowledge distillation—on most benchmarks, it consistently exceeds the performance of other foundation models. A key finding is that after applying reinforcement learning (RL), **our model outperforms DeepSeek-R1 on both RTLLM-v1.1 and RTLLM-v2**, underscoring the significant efficacy of the RL phase. The underwhelming results of other foundation models, such as Qwen2.5-Coder-Instruct and DeepSeek-R1-Distill-Qwen, highlight the limited exposure to Verilog data during their pre-training and instruction-tuning stages. This is further evidenced by the observation that distilling general-purpose knowledge from large models (e.g., in mathematics and software code) fails to enhance the Verilog capabilities of smaller models.

**Reinforcement learning significantly improves model performance.** Compared with CodeV-R1-7B-Distill, our RL model CodeV-R1-7B shows a noticeable improvement on almost all benchmarks. Especially on the RTLLM benchmark, the reinforcement learning process results in an improvement of over 10 % for the pass@1 score. This indicates great potential of RL for Verilog code generation and showcases the robustness of our testbench in providing reliable functional correctness rewards.

## 3.3 Additional Experiments

### 3.3.1 Test-Time Scaling

Test-time scaling is an important ability of reasoning LLMs [24]. To verify the test-time scaling ability of our CodeV-R1-7B, we take the RTLLM v1.1 dataset as an example and evaluate the accuracy of our model and DeepSeek-R1 under varying response length budgets. Formally, we force the response

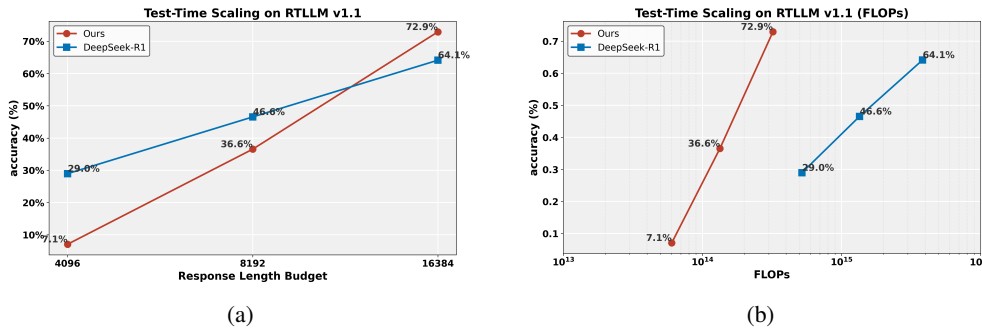

(a)                                              (b)

Figure 2: **Test-time scaling on RTLLM v1.1.** Figure (a) shows response length against accuracy, while Figure (b) shows FLOPs against accuracy. FLOPs are estimated according to model architecture.

length of both models to be smaller than certain thresholds (4096, 8192 and 16384 tokens), and plot the corresponding results in Figure 2a. To ensure fair comparison, we also normalized FLOPs consumption at each response length, as shown in Figure 2b.

Both models' accuracy improves considerably as the response length budget increases from 4096 to 16384. CodeV-R1-7B's accuracy rises from 7.1% to 72.9%, outperforming DeepSeek-R1 (29.0% → 64.1%). When evaluated in terms of FLOPs efficiency, CodeV-R1-7B demonstrated superior computational economy, delivering higher accuracy per unit of computation compared to DeepSeek-R1. These results underscore CodeV-R1-7B's exceptional test-time scaling efficiency, showcasing its ability to leverage longer contexts more effectively than DeepSeek-R1 while consuming fewer computational resources on the RTLLM v1.1 benchmark.

### 3.3.2   Equivalence Checking and Difficulty Filtering Improves RL Training

To explore whether equivalence checking and difficulty filtering improve RL dataset quality, we conduct an ablation study by constructing two additional datasets.

Our original RL dataset contains 3.1K problems where DeepSeek-R1 responses pass the equivalence checking, while both Qwen2.5-Coder-7B-Instruct and Qwen2.5-Coder-32B-Instruct fail across five sampling attempts. To conduct difficulty ablation, we introduce a **dataset without difficulty filtering** containing 16K problems, where we additionally include samples where Qwen2.5 models succeed in some attempts under our testbench. To conduct reference code correctness ablation, we introduce a **dataset without round-trip equivalence checking** containing 14K samples, where we treat DeepSeek-R1 outputs as pseudo-golden code. We select cases where Qwen2.5 models fail to match this pseudo-golden code in five attempts to control difficulty. To avoid time waste, we filter the problems where CodeV-R1-7B-Distill has a 100% pass rate under our testbench in five attempts.

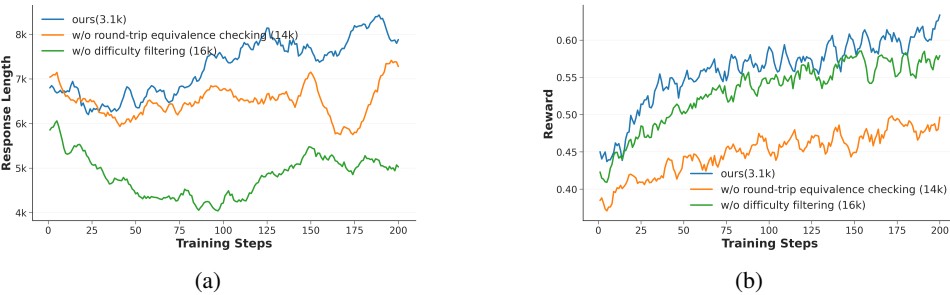

(a)                                              (b)

Figure 3: **Train-time scale up on some key metrics.** Figure (a) tracks response length, whereas Figure (b) presents the corresponding trend for reward.

We perform reinforcement learning using CodeV-R1-7B-Distill on the three aforementioned datasets, employing identical training parameters. Key metrics observed during these training processes are presented in Figure 3. Inspection of Figure 3a reveals distinct trends in response length during training. Utilizing the original RL dataset leads to a noticeable subsequent increase in response length, whereas the training dataset without difficulty filtering leads to a segment of response decrease.

This suggests that even when initial responses are relatively long, incorporating more challenging samples during reinforcement learning facilitates further steady growth in response length. Figure 3b illustrates that the pseudo-golden dataset consistently exhibits notably lower reward throughout the training process compared to our original RL dataset. This underscores the critical role of golden code accuracy during reinforcement learning.

### 3.3.3 Acceleration via Adaptive DAPO

To quantitatively demonstrate the acceleration achieved by our adaptive DAPO algorithm, we provide a comparison of time usage in Figure 4a. The plots reveal a notable increase in the time per RL step in baseline DAPO training around step 150. This performance degradation in the baseline is attributed to its fixed generation batch size, which becomes insufficient to yield enough samples for a complete training batch as training progresses. In contrast, our adaptive DAPO effectively mitigates this issue. It dynamically adjusts and increases the generation batch size across steps. In addition, when a generation attempt does not produce sufficient valid samples for a training batch, the algorithm recalculates the required remaining batch size. In Figure 4b, we provide the average speedup of adaptive DAPO, along with a breakdown of performance before and after step 150. Notably, the time reduction after step 150 is significantly more pronounced—the speedup factor reaches 1.44 after step 150, compared to 1.04 before step 150. This disparity highlights the critical benefit of eliminating sampling more than once. After applying adaptive DAPO, the final speedup factor reaches 1.25×.

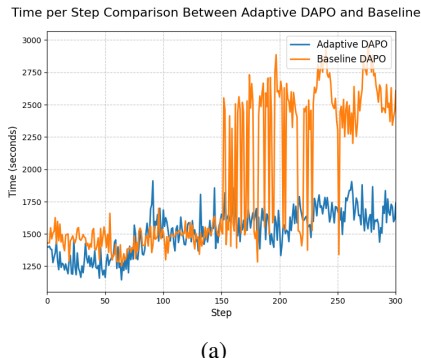 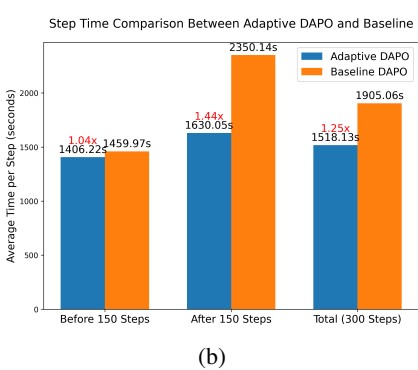

|        (a)         |        (b)         |

Figure 4: **Time comparison between adaptive DAPO and baseline DAPO. (a)**: Comparison of RL training time per step. **(b)**: Acceleration ratio between adaptive DAPO and baseline DAPO, breakdown by step (whether before 150).

### 3.3.4 Testbench Performance Evaluation

We evaluate our auto-testbench generation framework against a DeepSeek-V3-generated testbench, both taking the Verilog code from GitHub as the golden reference. We conduct two key tests:

**Correctness classification test.** We assess whether the testbenches might misclassify correct code as incorrect. To do this, we validate with "golden vs. golden" inputs (i.e., comparing the golden code against itself). The expected outcome is 100% correct classification. Our method misclassifies only 0.3% of cases (due to problems of Icarus Verilog simulation and randomization issues) — a 96.1% reduction in false negatives compared to the DeepSeek-V3-generated testbench (7.6%).

**Fuzzing test for sequential circuits.** Second, we perform a fuzzing test on sequential circuits by instructing DeepSeek-V3 to inject subtle errors into the golden code. The goal is to measure how effectively each testbench detects these mistakes. Our testbench detects 65% of the injected errors, demonstrating a 62.5% relative improvement in detection rate over the DeepSeek-V3 testbench (40%) and indicating fewer false positives.

## 4 Related Work

### 4.1 Large Language Models for Reasoning

The OpenAI-o1 [26] series is the first closed-source model trained with large-scale reinforcement learning to perform reasoning through CoT. Inspired by its powerful and effective reinforcement

learning training paradigm, QwQ [36], DeepSeek-R1 [3], and Kimi k1.5 [34] have all adopted and improved upon its approach, achieving promising results. Limited by computational resources, open-source communities have actively explored low-cost approaches to replicate o1-like reasoning models. Some efforts have focused on distilling the powerful closed-source reasoning models [8, 23, 24, 35, 43]. while others have also explored training reasoning models using reinforcement learning [7, 10, 18, 20, 21, 27, 40, 46].

The main difference between CodeV-R1-7B and the aforementioned reasoning models lies in its focus on hardware description language code generation, which poses unique challenges due to **verification difficulty** and **limited data quality**. In contrast, prior works primarily specialised in domains such as mathematics, which benefit from easily verifiable numerical outputs and rich open-source datasets.

## 4.2 Large Language Models for Verilog Code Generation

With the development of large language models, specialised code generation models for hardware description languages also receive widespread attention. Many prior works [2, 15, 16, 45, 48] focus on Verilog instruction-tuning data creation **without a strict correctness evaluation**. Most works have a syntax check in constructing instruction-response pairs: RTLCoder [16] and CodeV [48] add syntax checks when constructing supervised fine-tuning (SFT) datasets with closed-source LLMs. BetterV [45] maps code across languages using Verilog syntax constraints, while OriGen [2] leverages compiler feedback to eliminate syntax errors. For functional correctness, to date, only CraftRTL's correct-by-construction approach [15] ensures functional correspondence between instruction and response through formal verification. However, its applicability remains restricted to Karnaugh maps and finite-state machines, a narrow subset of Verilog design challenges.

This verification bottleneck shifts to the model optimization stage. Specifically, reinforcement learning with rule-based rewards attempts to address functional correctness by relying on testbenches for reward calculation. However, this strategy is undermined by the fact that current testbench generation paradigms suffer from two systemic flaws: (1) **Unverified validation frameworks**: For example, VeriPrefer [37] optimizes testbench coverage, but its testbenches themselves may be flawed, sometimes failing to pass the reference code they were designed to verify. ReasoningV [29] co-generates code and testbenches via DeepSeek-R1, inheriting the model's hallucination risks. (2) **Cost-prohibitive iteration**: AutoBench [30] and CorrectBench [31] employ multi-stage LLM workflows, where each self-correction cycle incurs escalating computational costs and latency, directly conflicting with RL's demand for rapid, low-cost reward feedback.

Unlike prior work, we apply Verilog functional verification with auto-generated equivalence checking (see Section 2.1), providing a robust foundation for both data curation and reinforcement learning.

## 5 Conclusion

In this paper, we propose CodeV-R1, a unified RLVR framework designed for training Verilog generation LLMs. This framework first distills data with reasoning patterns and then applies reinforcement learning on high-quality data curated by an automated testbench generation framework. The model trained via this framework, CodeV-R1-7B, achieves outstanding performance on Verilog generation benchmarks like VerilogEval and RTLLM, matching or even surpassing DeepSeek-R1, which demonstrates the effectiveness of the automated testbench generation and the two-stage training paradigm. Notably, to the best of our knowledge, CodeV-R1-7B is the first Verilog-domain LLM to demonstrate strong reasoning capability showing consistent test-time scaling behavior. A series of analytical experiments further highlights the powerful impact of CodeV-R1 framework in enhancing data quality and further unlocking the Verilog code generation capabilities of LLMs through reasoning.

## Acknowledgements

This work is partially supported by the Strategic Priority Research Program of the Chinese Academy of Sciences (Grants No.XDB0660300, XDB0660301, XDB0660302), the NSF of China (Grants No.62341411, 62222214, 62525203, U22A2028, 6240073476), CAS Project for Young Scientists in Basic Research (YSBR-029) and Youth Innovation Promotion Association CAS.

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

# A  Method Details

## A.1  Proof and Further Explanation of Theorem 2.1

*Proof.* Observe that the sequence $Y \to X \to Y'$ forms a Markov chain. By the Data Processing Inequality (DPI),

$$I(Y;Y') \leq I(Y;X).$$

Under the assumption that $E_{Y,Y'}$ holds almost surely, we have $H(Y \mid Y') = 0$, and thus

$$I(Y;Y') = H(Y) - H(Y \mid Y') = H(Y).$$

It follows that

$$H(Y) = I(Y;Y') \leq I(Y;X) \leq H(Y) \implies I(Y;X) = H(Y) \implies H(Y \mid X) = 0,$$

meaning $Y$ is determined by $X$ almost surely and hence $E_{Y,X}$ holds.

Next, since $H(Y \mid X) = 0$ implies $H(X) = H(Y)$ and $I(X;Y') \leq H(X)$, a failure of $E_{X,Y'}$ under the NLCDE assumption would force

$$I(X;Y') < H(X) = H(Y),$$

contradicting $I(Y;Y') = H(Y)$. Therefore, $E_{X,Y'}$ must also hold almost surely. Combining these two results gives

$$E_{Y,X} \wedge E_{X,Y'} \quad \text{holds almost surely.}$$

$\square$

**Remark**: The need for NLCDE in Theorem 2.1 arises because $E_{X,Y} \Rightarrow H(Y \mid X) = 0$, but $H(Y \mid X) = 0 \nRightarrow E_{X,Y}$. A counterexample is when $X$ and $Y$ are incorrectly matched with probability one. To be more specific (though not fully rigorous, just to aid understanding), if the NL-to-code model wrongly transforms $A$ (e.g., "design a multiplier") in the NL domain to $B$ (e.g., "design an adder") in the code domain, and transforms $B$ in the NL domain to $A$ in the code domain, while the code-to-NL model maps $A$ in the code domain to $B$ in the NL domain and $B$ in the code domain to $A$ in the NL domain, then $H(Y \mid Y') = 0$ can hold without $E_{Y,Y'}$. Thus, the NLCDE assumption is necessary to resolve this.

**Further explanation**: Here we re-emphasize some critical points of this theorem:

1. **Functional Identity**: The theory is built upon the space of code functions ($\mathcal{F}$) and natural language descriptions ($\mathcal{L}$). Different code snippets / NL descriptions that implement / describe the same function (e.g., the same RTL module) are **considered identical** within $\mathcal{F}$ / $\mathcal{L}$.

2. **Interpretation of Determinism**: The assumption of deterministic mappings ($M_1 : \mathcal{F} \to \mathcal{L}$ and $M_2 : \mathcal{L} \to \mathcal{F}$) models the high functional consistency (not textual uniformity) achieved by capable LLMs. As model capability increases, the mapping from a precise NL description to a core code function becomes increasingly stable (probability converges to 1).

3. **Theoretical Relevance**: Theorem 2.1 establishes that, under these idealized conditions, the functional equivalence (of $Y$ and $Y'$) after the round-trip process ($Y \to X \to Y'$) can guarantee the correctness of both problem summarization ($Y \to X$) and code generation ($X \to Y'$). This provides a foundational principle explaining why our synthesis loop can bootstrap high-quality, self-consistent data, a principle that is strongly supported by our empirical outcomes.

## A.2  Algorithm Description of Adaptive DAPO

In this section, we provide the algorithm description of adaptive DAPO in Algorithm 1. In this algorithm, one epoch means going through the whole training dataset, while one step is to collect enough samples and update the model parameters like standard DAPO [44]. Note that we achieve the dynamic batch size by two granularities: First, we use a step-level ratio $r_{valid}$ to control the generation batch size $b_{gen}$. Second, if one generation does not provide enough samples for training, we use another inner-step-level ratio $r_{step}$ to control the generation batch size for the remaining samples.

**Algorithm 1** Adaptive DAPO

**Require:** Training batch size $b_{train}$, dataset $\mathcal{D}$
**Ensure:** Updated $r_{valid}$ and filtered problem pool
  Initialize $r_{valid} \leftarrow 1$
  **for** epoch $= 1, 2, \dots$ **do**
    Shuffle $\mathcal{D}$                                                  *(Epoch reset)*
    $N_{total} \leftarrow |\mathcal{D}|, N_{consumed} \leftarrow 0$
    **while** $N_{consumed} < N_{total}$ **do**                         *(Process epoch)*
      $\Sigma b_{gen} \leftarrow 0, n_{valid} \leftarrow 0, r_{step} \leftarrow r_{valid}$
      **while** $n_{valid} < b_{train}$ **do**
        $b_{remain} \leftarrow b_{train} - n_{valid}$
        $b_{ge} \leftarrow \lceil b_{remain}/r_{step} \rceil$                             *(Dynamic batch)*
        $\mathcal{D}' \leftarrow \mathcal{D}[N_{consumed} : \min(N_{consumed} + b_{ge}, N_{total})]$
        Generate $b_{ge}$ samples from $\mathcal{D}'$
        Update counters: $n_{valid} \leftarrow n_{valid} + v_{new}, \Sigma b_{gen} \leftarrow \Sigma b_{gen} + b_{ge}$
        $r_{step} \leftarrow \min \left( r_{step}, \frac{n_{valid}}{\Sigma b_{gen}} \right)$
      **end while**
      Update ratio: $r_{valid} \leftarrow \min \left( r_{valid}, \frac{n_{valid}}{\Sigma b_{gen}} \right)$
      Train DAPO with $b_{train}$ valid samples                 *(RL step)*
    **end while**
  **end for**

# B   Parameter Setting

The full parameter setting during the SFT (distillation) stage is shown in Table 3, while the full parameter setting during the RL stage is shown in Table 4. During testing, we use a max context length of 16384 and a temperature of 1.0. We set top_p to 1.0 for VerilogEval and 0.95 for RTLLM.

For RL, the generation batch size in Table 4 corresponds to train_batch_size in verl [32], and the training batch size corresponds to ppo_mini_batch_size in verl. A generation batch size of 128 and training batch size of 64 (with a rollout number of 16) means first generating $128 \times 16$ samples for 128 problems and updating two times, each with $64 \times 16$ samples, during one RL step. Meanwhile, the clip ratio(high), clip ratio(low), overlong penalty factor, and overlong response length in Table 4 are introduced by DAPO. Here, the max train response length in Table 4 corresponds to $L_{max}$ in DAPO, and the overlong response length corresponds to $L_{cache}$. The overlong penalty in DAPO $P_{length}(y)$ (where y is response length) is defined as:

$$P_{length}(y) = \begin{cases} 0, & |y| \le L_{max} - L_{cache} \\ -\frac{|y| - (L_{max} - L_{cache})}{L_{cache}}, & L_{max} - L_{cache} < |y| \le L_{max} \\ -1, & L_{max} < |y|, \end{cases} \tag{2}$$

which is added to the {0, 1} reward.

Table 3: SFT Parameter Setting.

| Parameter Category | Parameter Name | Value | Parameter Name | Value |
|---|---|---|---|---|
| Training Mode | Finetuning Type | Full Parameter | Deepspeed | Zero3 |
| Optimization & Scheduling | Epochs | 6 | Learning Rate (LR) | $1 \times 10^{-5}$ |
| | Batch Size | 64 | Optimizer | AdamW |
| | LR Scheduler | Cosine Decay | LR Warmup Ratio | 0.03 |
| | Numerical Precision | BF16 | | |
| Context & Data Handling | Max Context Length | 16384 | Packing | True |

Table 4: RL Parameter Setting.

| Parameter Category | Parameter Name | Value | Parameter Name | Value |
|---|---|---|---|---|
| Batch Size Related | Generation Batch Size | 128 | Training Batch Size | 64 |
| | Dynamic Batch Size | True | | |
| Rollout Configuration | Rollout Number | 16 | Rollout Temperature | 1.0 |
| | Rollout Engine | VLLM | Rollout GPU Memory Utilization | 0.8 |
| Optimization & Regularization | Learning Rate | $1 \times 10^{-6}$ | Weight Decay | 0.0 |
| | KL Coefficient | 0.0 | KL Loss Coefficient | 0.0 |
| Clipping & Penalty | Clip Ratio (High) | 0.28 | Clip Ratio (Low) | 0.2 |
| | Overlong Penalty Factor | 1.0 | | |
| Length Control | Max Train Response Length (Full) | 16384 | Overlong Response Length | 1024 |
| | Max Generate Response Length | 32768 | | |
| Computation & Memory Optimization | Gradient Clip | 0.5 | Gradient Checkpointing | True |
| | Use Liger Kernel | True | VLLM Enforce Eager | False |
| | Tensor Parallel Size | 4 | | |
| Distributed Training Configuration | Number of Nodes | 2 | GPUs per Node | 8 |
| Data Processing | Remove Padding | True | Token Level Loss | True |
| FSDP Related | FSDP Optimizer Offload | False | FSDP Parameter Offload | False |

# C  Additional Statistics and Analysis

## C.1  Benchmark Comparison

Since there is a notable performance gain difference (especially for the RL phase) of our method between VerilogEval and RTLLM, we provide a deeper analysis of this phenomenon in this section. Given that RTLLM's performance gains stem mainly from reinforcement learning, we focus on distribution differences between our RL dataset and the two benchmarks.

**Distribution similarity to RTLLM**: We run the instructor-embedding model [33] for the golden code in our RL dataset, RTLLM (v2), and VerilogEval (v2 spec-to-RTL), then generate a t-SNE distribution plot [22] in Figure 5. This plot revealed that our RL dataset aligns closely with RTLLM but diverges from VerilogEval for both problems and solutions.

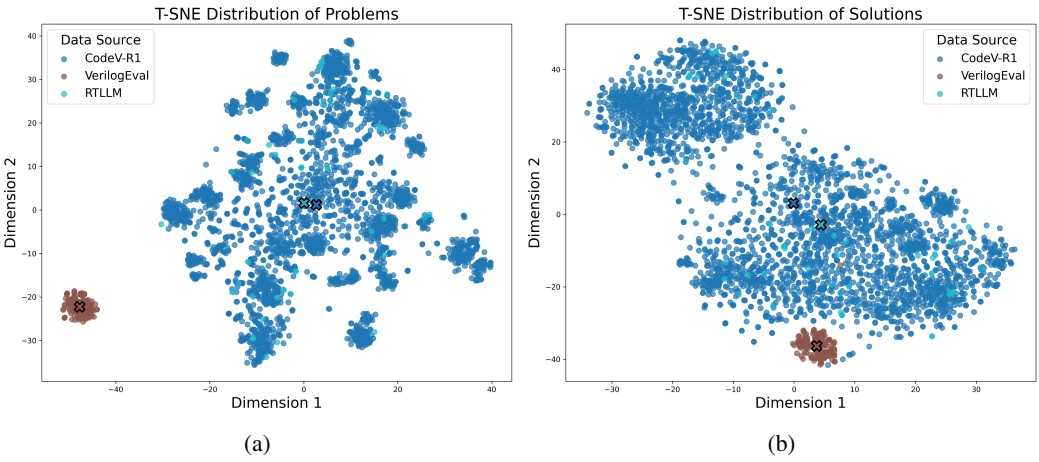

Figure 5: **T-SNE distribution** of CodeV-R1 RL dataset, RTLLM (v2), and VerilogEval (v2 spec-to-RTL). **Left**: Problem (NL) distribution; **Right**: Solution (code) distribution.

Table 5: Centroid Distance and Similarity between Our RL Dataset and Benchmarks

| | RTLLM v2 | VerilogEval v2 |
|---|---|---|
| Euclidean Distance | 0.1150 | 0.2903 |
| Cosine Similarity | 0.9926 | 0.9547 |

We also show cosine similarity and Euclidean distance metrics for the embedding centroids between our RL dataset and the benchmarks in Table 5. Our RL dataset's embedding centroid shows significantly smaller Euclidean distance and closer-to-1 (maximum value) cosine similarity with RTLLM

than with VerilogEval. Importantly, this reflects only embedding centroid relationships, not data homogenization or overfitting to RTLLM.

**Problem type difference**: Additionally, we conducted a detailed case study across both benchmarks and identified that one reason for this performance gain difference is VerilogEval's heavier use of table/graph-based problems, where our model underperforms significantly. Table 6 presents a comparison of our model's accuracy against DeepSeek-R1 on these problem types, including their ratios within benchmarks:

Table 6: Accuracy Comparison on Table/Graph Problems Across Benchmarks

|  | Tables/Graphs (Ours) | Tables/Graphs (DS-R1) |
| --- | --- | --- |
| VerilogEval v2 (Ratio) | 29.49% | 29.49% |
| VerilogEval v2 (Accuracy) | 55.43% | 78.04% |
| RTLLM v2 (Ratio) | 0% | 0% |
| RTLLM v2 (Accuracy) | N/A | N/A |

This points to a key improvement direction: incorporating more table/graph-specific instruction-response pairs (e.g., KMap, FSM, waveform data as in CraftRTL [15]) into our training dataset.

**Benchmark complexity comparison**: Additionally, we provide the token count comparison, serving as a proxy for complexity, for these two benchmarks. Table 7 provides the average number of lines and tokens (comments and blank lines removed) for VerilogEval and RTLLM. From these results, we can clearly see that RTLLM is generally more complex than VerilogEval.

Table 7: Code Length Comparison between VerilogEval and RTLLM Datasets

| Dataset | Average Lines | Average Tokens |
| --- | --- | --- |
| VerilogEval v1 Machine | 13.89 | 93.85 |
| VerilogEval v1 Human | 15.82 | 117.35 |
| VerilogEval v2 Code Completion | 16.10 | 121.84 |
| VerilogEval v2 Spec to RTL | 16.12 | 122.31 |
| RTLLM v1.1 | 56.52 | 470.48 |
| RTLLM v2.0 | 46.30 | 403.80 |

## C.2 Additional Benchmark Statistics

In this section, we take a close look at the mistake type on VerilogEval v2 and the pass@k metrics of different task types on RTLLM v2.

Table 8: Comparison of Error Types for VerilogEval v2.

| Model | Compiler Errors | | | | | | | | | Runtime Errors | | | |
| --- | --- | --- | --- | --- | --- | --- | --- | --- | --- | --- | --- | --- | --- |
|  | C | S | w | m | p | e | n | c | Total | R | T | r | Total |
| CodeV-R1-7B-Distill | 69 | 107 | 108 | 27 | 20 | 1 | 0 | 0 | 332 | 1699 | 107 | 19 | 1825 |
| CodeV-R1-7B | 63 | 38 | 46 | 22 | 3 | 0 | 1 | 0 | 173 | 1610 | 114 | 1 | 1725 |
| DeepSeek-R1-671B | 37 | 59 | 47 | 0 | 0 | 1 | 0 | 0 | 144 | 1096 | 110 | 5 | 1211 |

$^*$ Error type explanation: C – General Compiler Error; S – Syntax Error; w – Reg Declared as Wire; m – Module Missing; p – Unable to Bind Wire/Reg; e – Explicit Cast Required; n – Sensitivity Problem; c – Unable to Bind Wire/Reg 'clk'; R – General Runtime Error; T – Timeout. r – Reset Issue;

**Mistake type:** As shown in Table 8, our RL training notably reduces error rates, particularly for compiler errors. CodeV-R1-7B achieves a 48% reduction in total compiler errors compared to CodeV-R1-7B-Distill (from 332 to 173), with the most pronounced improvements in syntax errors (S, reduced by 65% from 107 to 38) and wire declaration issues (w, down 57% from 108 to 46). Notably, our CodeV-R1-7B has a remarkably fewer syntax error (38) compared to DeepSeek-R1 (59) and fewer reset issues (r) (1 vs 5). Even so, our CodeV-R1-7B still has limitations. For instance, the number of general runtime errors (R) is still notably higher than DeepSeek-R1. This might stem from the RL training data not being suitable for VerilogEval (unlike the great improvement on RTLLM).

Table 9: Performance Across Different Module Categories on RTLLM v2.

| Model | Arithmetic (%) | | Control (%) | | Memory (%) | | Miscellaneous (%) | |
|---|---|---|---|---|---|---|---|---|
| | pass@1 | pass@5 | pass@1 | pass@5 | pass@1 | pass@5 | pass@1 | pass@5 |
| CodeV-R1-7B-Distill | 69.47 | 89.19 | 74.17 | 83.06 | 43.57 | 56.69 | 46.94 | 59.72 |
| CodeV-R1-7B | 83.68 | 91.66 | 80.00 | 83.33 | 51.43 | 63.30 | 57.78 | 72.40 |
| DeepSeek-R1-671B | 76.58 | 90.65 | 83.33 | 83.33 | 57.14 | 60.71 | 52.50 | 67.72 |

**Accuracy among task types:** Table 9 demonstrates the comparative performance across module categories, where CodeV-R1-7B shows consistent improvements over CodeV-R1-7B-Distill while maintaining competitive results against the larger DeepSeek-R1. Notably, CodeV-R1-7B achieves superior pass@1 rates in all categories over CodeV-R1-7B-Distill, with particularly strong gains in arithmetic modules (83.68% vs 69.47%) and miscellaneous modules (57.78% vs 46.94%). It also surpasses DeepSeek-R1 in these two categories. Compared with the training dataset classification provided in Figure 6, these two categories occupy a larger portion (arithmetic and others). This observation suggests that augmenting the training set with high-quality RL data for currently under-performing categories (particularly Memory and Control modules) could be a productive direction for future model improvement.

## C.3 Training Dataset Statistics

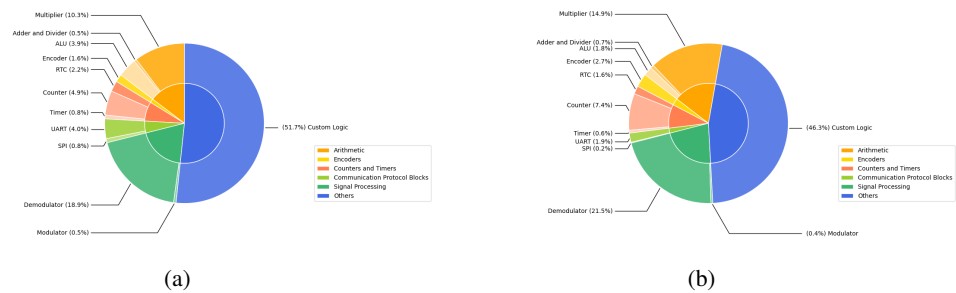

(a)                                          (b)

Figure 6: **Problem category distribution. Left**: SFT dataset; **Right**: RL dataset.

Figure 6 presents the category distribution of our 87K SFT and 3.1K RL training datasets (categorized using both questions and answers). While both datasets show comparable distributions, the RL dataset has fewer unclassified problems.

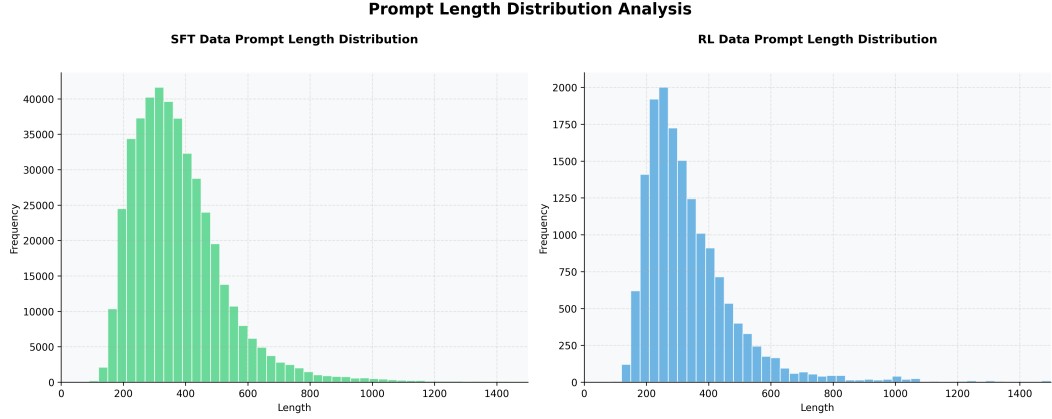

Figure 7: **Prompt length distribution. Left**: SFT dataset; **Right**: RL dataset.

Figure 7 illustrates the prompt length distribution (in tokens) for our 87K SFT and 3.1K RL training datasets, both clipped to a maximum prompt length of 1500 tokens. The figure reveals a sharper distribution for the RL data, indicating shorter and lower-variance prompt lengths compared to the

SFT data. To quantify this observation, we calculated the following statistics: The average length of SFT data is 377.81 with a standard deviation of 161.30, while the average length of RL data is 336.67 with a standard deviation of 153.88. These statistics align with the visual trends in the figure.

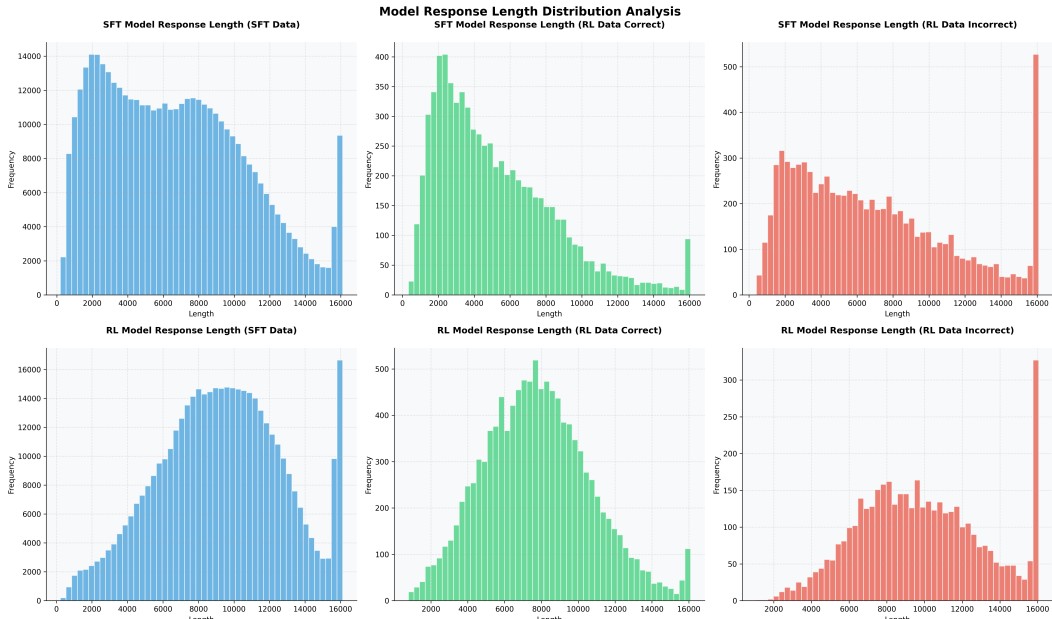

Figure 8: **Response length distribution. Left**: SFT dataset; **Middle**: Correct samples in RL dataset; **Right**: Incorrect samples in RL dataset.

Figure 8 depicts the response length distributions (in tokens) for CodeV-R1-7B-Distill and CodeV-R1-7B. Note that the maximum context length—the sum of prompt length and response length—is capped at 16384 tokens. Consequently, when responses are truncated, their recorded length is 16384 tokens minus the prompt length, resulting in a somewhat scattered distribution (manifested as the two rightmost bars, instead of one, become longer in the distribution plot). The response length for CodeV-R1-7B exhibits an evident right shift, indicating longer responses after reinforcement learning. Additionally, CodeV-R1-7B's response distribution is more symmetric compared to the left-skewed distribution of CodeV-R1-7B-Distill. The underlying cause of this discrepancy warrants further investigation. We observe that incorrect samples are significantly longer, with a substantial proportion exceeding the length threshold. Even excluding these overlong samples, incorrect responses remain longer, characterized by a higher peak value (CodeV-R1-7B) or a slower post-peak decline (CodeV-R1-7B-Distill). An intriguing phenomenon is that CodeV-R1-7B has a lower overlong ratio on the RL dataset but a higher ratio on the SFT dataset. This may arise from overfitting the overlong penalty during RL, while CodeV-R1-7B's tendency to generate longer responses increases overlong instances on the SFT dataset.

## C.4   Agent Ability Analysis

This section presents supplementary experiments on agentic integration capabilities for our CodeV-R1-7B with the MAGE [49] framework. We identify two fundamental distinctions between MAGE and our approach: (1) MAGE utilizes golden testbenches for verification while CodeV-R1 operates without them, and (2) MAGE's performance exhibits strong dependency on both prompt engineering strategies and the underlying model capabilities.

Hence, we employ a targeted experimental design to isolate and evaluate CodeV-R1's efficacy within a multi-agent system. Specifically, we replaced only the RTL generation agent in MAGE with CodeV-R1—as this role best aligns with CodeV-R1's core capabilities—while maintaining the other agents based on Claude-3.5-Sonnet.

The experimental results in Table 10 demonstrate that CodeV-R1 achieves competitive performance (94.87% Pass@1) compared to the best-performing model (Claude-3.5-Sonnet at 95.51%), while sub-

stantially outperforming other baseline models. This indicates CodeV-R1's potential for integration into agent workflows and suggests promising directions for developing end-to-end agent capabilities in future work.

Table 10: Agent Performance Comparison in MAGE Framework

| Model | Pass@1 |
|---|---|
| Deepseek-R1-0528 | 80.77% |
| DeepSeek-V3 | 88.46% |
| GPT-4o-240806 | 76.28% |
| Claude-3.5-Sonnet | 95.51% |
| CodeV-R1(with Claude) | 94.87% |

# D   Case Study

**Comparison Between CodeV-R1-7B and DeepSeek-R1**: In this section, we first present a case study on a specific Verilog problem in VerilogEval-v2 to illustrate the advantage of CodeV-R1-7B over DeepSeek-R1 in this problem. The problem, reasoning chains, and results are shown in the blocks below.

Both Deepseek-R1 and our model comprehended the problem and engaged in reasoning and reflection, but only our model accurately implemented the logical function defined by the Karnaugh map, while R1's answer executed an incorrect logical function. The key distinction lies in our model's use of the declaration *input [4:1] x*, which directly corresponds to the notation of the Karnaugh map in the problem description. In contrast, R1 employed *input [3:0] x*, leading to confusion in subsequent reasoning. Although our model was initially challenged by this unconventional declaration, it ultimately chose the correct declaration through reflection, avoiding potential index mapping ambiguities. This resulted in significantly clearer code that is less prone to errors.

---

**Problem Description**

**Question:**

Question: I would like you to implement a module named TopModule with the following interface. All input and output ports are one bit unless otherwise specified.
- input x (4 bits) - output f
The module should implement the function f shown in the Karnaugh map below. d is don't-care, which means you may choose to output whatever value is convenient.
```
          x[1]x[2]
x[3]x[4] 00 01 11 10
  00    | d | 0 | d | d |
  01    | 0 | d | 1 | 0 |
  11    | 1 | 1 | d | d |
  10    | 1 | 1 | 0 | d |
```
Answer:

---

Okay, so I need to write a Verilog module called TopModule that takes a 4-bit input x and outputs a single bit f based on the given Karnaugh map. Let me start by understanding the problem step by step.

...

Wait, the K-map's rows are x3x4, columns x1x2. Let's see. For example, the first row is x3x4 = 00. So the row is determined by the bits x3 and x4. Similarly, columns are determined by x1 and x2. So each cell in the K-map is determined by the combination of x3, x4, x1, x2. So the four input bits are x1, x2, x3, x4. But how are these mapped to the input x in Verilog? The input x is a 4-bit vector. Let's assume that x is declared as [3:0] x, where x[3] is the first bit (MSB) and x[0] is the last. So perhaps the bits are arranged as x[3] = x1, x[2] = x2, x[1] = x3, x[0] = x4. But this is a guess. Alternatively, the problem might have a typo, and the rows are x[2]x[3], and columns x[0]x[1]. But given the problem statement, I need to proceed with the given labels.

...

Now, the goal is to cover all the 1's and 0's with the largest possible groups, using don't-cares as either 1 or 0 to maximize grouping.

First, let's identify the 1's and 0's.

Looking for 1's:

In row 01 (x3=0, x2=1), column 11 (x1=1, x0=1) → 1.

In row 11 (x3=1, x2=1), columns 00 and 01 → 1's.

In row 10 (x3=1, x2=0), columns 00, 01 → 1's.

In row 10, column 11 → 0. Wait, no: row 10, column 11 (x1=1, x0=1) is 0. So that's a 0.

The 0's are:

Row 00, column 01 → 0.

Row 01, columns 00, 10 → 0.

Row 10, column 11 → 0.

The don't-cares can be used to simplify the expression. So for f, we can choose to treat the don't-cares as 1 or 0 to form the largest possible groups.

...

Therefore, the Verilog code for the module would be:

module TopModule ( input [3:0] x, output f );

```
    assign f = (x[3] & ~x[1]) | (~x[3] & x[1] & x[0]);
```

endmodule

But let me double-check this.

Another way to check is to test some values.

Test case 1: x3=1, x2=1, x1=0, x0=0 → f should be 1.

In the K-map: row x3x2=11 (row 11), column x1x0=00 → 1. Our expression: $(1 \& \tilde{0}) | (0 \& 0 \& 0) \to 1$ | 0 = 1. Correct.

Test case 2: x3=1, x2=0, x1=0, x0=1 → row 10, column 01 → 1. Expression: (1 & 1) | (0 & ... ) → 1. Correct.

...

Test case 6: x3=0, x2=0, x1=1, x0=1 → row 00, column 11 (don't-care). Our expression sets this to 1 via x̃3 x1 x0. So f=1 here. Which is acceptable as a don't-care.

Thus, the Verilog code should correctly implement the K-map.

...
**Solution**
```verilog
module TopModule (
    input [3:0] x,
    output f
);

    assign f = (x[3] & ~x[1]) | (~x[3] & x[1] & x[0]);

endmodule
```
...

<think>Okay, let's try to tackle this Verilog problem. The goal is to create a module called TopModule with a 4-bit input x and a single-bit output f. The function is determined by the Karnaugh map provided.
...
Let me try to find the logic.
Looking for the minterms where f=1, considering don't cares as 1s.
Wait the f=1 in the K-map.
Looking across the K-map, the entries that have 1 are:
In x3x4=11 row:
Columns x1x2 when:
x1x2=00 (f=1) and x1x2=01 (f=1).
In x3x4=10 row:
x1x2=00 and 01 (so x3x4=10, x1x2=00 and 01 → which gives x1=0, x2=0 and x1=0, x2=1. So x3x4 can be 10 and x1x2=00 and 01.
Also:
Row x3x4=01 and x1x2=11: f=1.
Row x3x4=01's column x1x2=11 (third column) gives 1.
...
The variables are x[1], x[2], x[3], x[4]. Wait the input is x as four bits. Let's suppose the four bits are x[3:0], but the code might have x[3], x[2], x[1], x[0], but the naming here may vary.
Wait the K-map labels the x as x[1] to x[4]. So in Verilog, if input is declared as [4:1], which is a standard way.
But in Verilog, arrays are usually declared with [width-1 : 0], but the actual variable name is [high:low].
For example, input [4:1] x would have indices x[4], x[3], x[2], x[1], x[0].
But in the Karnaugh map, the rows are x3 and x4, columns are x1 and x2.
But perhaps in the Verilog code, the variables are named x[1],x[2],x[3],x[4]. Let's assume the input is a four-bit vector named x, with each bit labeled x1, x2, x3, x4, such that:
input [4:1] x; // x[1], x[2], x[3], x[4]
But in Verilog, you can declare it as:
input [1:4] x; // but the syntax is different.
Alternatively, input [4:1] x; where x[1] is the first bit, etc.
But given the problem statement's variables as x[1] to x[4], the code can be written as:
module TopModule (input [4:1] x, output f);
Then, the variables are:
x[1], x[2], x[3], x[4].
...
So the final code would look like: module TopModule (input [4:1] x, output f);

assign f = (x[3] & ~x[1]) | (~x[3] & x[4] & x[1] & x[2]);

endmodule Wait wait, the variables in the second term are x3=0 (because x[3], x4=1 (x[4]), x1=1 (x[1]), x2=1 (x[2]).
Yes.
So that's correct.
...
</think>

<answer> ```verilog
module TopModule (input [4:1] x, output f);

    assign f = (x[3] & ~x[1]) | (~x[3] & x[4] & x[1] & x[2]);

endmodule
```</answer>

**Low-quality Data Example**: Below is an example of low-quality data in our dataset. The original code is just an empty module with some comments, while the summarized problem describes a module with an unsigned 32-bit divider. The problem and the code are inconsistent in this case.

## Original Code from GitHub

```verilog
module div_unsigned (
        aclr ,
        clock ,
        denom ,
        numer ,
        quotient ,
        remain );
        input      aclr;
        input      clock;
        input   [31:0]  denom;
        input   [31:0]  numer;
        output  [31:0]  quotient;
        output  [31:0]  remain;
endmodule
// Retrieval info: PRIVATE: new_diagram STRING "1"
// Retrieval info: LIBRARY: lpm lpm.lpm_components.all
// Retrieval info: CONSTANT: LPM_DREPRESENTATION STRING "UNSIGNED"
// Retrieval info: CONSTANT: LPM_HINT STRING "MAXIMIZE_SPEED=6,LPM_REMAINDERPOSITIVE=TRUE"
// Retrieval info: CONSTANT: LPM_NREPRESENTATION STRING "UNSIGNED"
// Retrieval info: CONSTANT: LPM_PIPELINE NUMERIC "11"
// Retrieval info: CONSTANT: LPM_TYPE STRING "LPM_DIVIDE"
// Retrieval info: CONSTANT: LPM_WIDTHD NUMERIC "32"
// Retrieval info: CONSTANT: LPM_WIDTHN NUMERIC "32"
// Retrieval info: USED_PORT: aclr 0 0 0 0 INPUT NODEFVAL "aclr"
// Retrieval info: USED_PORT: clock 0 0 0 0 INPUT NODEFVAL "clock"
// Retrieval info: USED_PORT: denom 0 0 32 0 INPUT NODEFVAL "denom[31..0]"
// Retrieval info: USED_PORT: numer 0 0 32 0 INPUT NODEFVAL "numer[31..0]"
// Retrieval info: USED_PORT: quotient 0 0 32 0 OUTPUT NODEFVAL "quotient[31..0]"
// Retrieval info: USED_PORT: remain 0 0 32 0 OUTPUT NODEFVAL "remain[31..0]"
// Retrieval info: CONNECT: @aclr 0 0 0 0 aclr 0 0 0 0
// Retrieval info: CONNECT: @clock 0 0 0 0 clock 0 0 0 0
// Retrieval info: CONNECT: @denom 0 0 32 0 denom 0 0 32 0
// Retrieval info: CONNECT: @numer 0 0 32 0 numer 0 0 32 0
// Retrieval info: CONNECT: quotient 0 0 32 0 @quotient 0 0 32 0
// Retrieval info: CONNECT: remain 0 0 32 0 @remain 0 0 32 0
// Retrieval info: GEN_FILE: TYPE_NORMAL div_unsigned.v TRUE
// Retrieval info: GEN_FILE: TYPE_NORMAL div_unsigned.inc TRUE
// Retrieval info: GEN_FILE: TYPE_NORMAL div_unsigned.cmp TRUE
// Retrieval info: GEN_FILE: TYPE_NORMAL div_unsigned.bsf TRUE
// Retrieval info: GEN_FILE: TYPE_NORMAL div_unsigned_inst.v TRUE
// Retrieval info: GEN_FILE: TYPE_NORMAL div_unsigned_bb.v TRUE
// Retrieval info: GEN_FILE: TYPE_NORMAL div_unsigned_syn.v TRUE
// Retrieval info: LIB_FILE: lpm
```

## Summarized Problem for The Code

You need to design a Verilog module that performs unsigned division on two 32-bit numbers. The module should have an asynchronous clear signal (aclr) to reset the division operation, and a clock signal (clock) to synchronize the process. The module should take two 32-bit inputs: the divisor (denom) and the dividend (numer). It should produce two 32-bit outputs: the quotient (quotient) and the remainder (remain). The module should be optimized for speed with an 11-stage pipeline and should ensure that the remainder is always positive. This Verilog module, named 'div_unsigned', has the interface designed as follows:

| Signal Name | Direction | Width | Description |
| --------- | ------- | ---- | --------------------------------- |
| 'aclr' | Input | 1 | Asynchronous clear signal to reset the module |
| 'clock' | Input | 1 | Clock signal to synchronize the division operation |
| 'denom' | Input | 32 | 32-bit unsigned divisor input |
| 'numer' | Input | 32 | 32-bit unsigned dividend input |
| 'quotient' | Output | 32 | 32-bit unsigned quotient output |
| 'remain' | Output | 32 | 32-bit unsigned remainder output, always positive |

# E   Prompts

Below shows the prompt for generating instructions given the GitHub codes by DeepSeek-V3.

## Prompt for Generating Questions Given Codes

### demonstration1
Please write a description about the code snippet and a problem such that the given code snippet is exactly the answer to this problem.
[The Code Snippet]
```verilog
module top_module(
        input ring,
        input vibrate_mode,
        output ringer,
        output motor
);
        assign ringer = ring & ~vibrate_mode;
        assign motor = ring & vibrate_mode;
endmodule
```
[Description]
This Verilog module is a top-level module that controls the ringer and motor of a device. It has four ports: two inputs (ring and vibrate_mode) and two outputs (ringer and motor). The ring input is a signal that is used to indicate when the device should be ringing or vibrating. The vibrate_mode input is a signal that indicates whether the device should be ringing or vibrating. The ringer output is a signal that is used to control the ringer of the device. It is set to 1 when the device should be ringing and 0 when it should not be ringing. The motor output is a signal that is used to control the motor of the device. It is set to 1 when the device should be vibrating and 0 when it should not be vibrating. The module uses two assign statements to control the ringer and motor outputs. The first assign statement sets the ringer output to the logical AND of the ring and the logical NOT of the vibrate_mode inputs. This means that the ringer output will be set to 1 when the ring input is 1 and the vibrate_mode input is 0, and it will be set to 0 otherwise. The second assign statement sets the motor output to the logical AND of the ring and the vibrate_mode inputs. This means that the motor output will be set to 1 when both the ring input and the vibrate_mode input are 1, and it will be set to 0 otherwise. This module is used to control the ringer and motor of a device based on the ring and vibrate_mode inputs. When the ring input is 1 and the vibrate_mode input is 0, the ringer output will be set to 1 and the motor output will be set to 0. When the ring input is 1 and the vibrate_mode input is 1, the ringer output will be set to 0 and the motor output will be set to 1.
[Problem]
Suppose you are designing a circuit to control a cellphone's ringer and vibration motor. Whenever the phone needs to ring from an incoming call (input ring), your circuit must either turn on the ringer (output ringer = 1) or the motor (output motor = 1), but not both. If the phone is in vibrate mode (input vibrate_mode = 1), turn on the motor. Otherwise, turn on the ringer. This Verilog module, named 'top_module', has the interface designed as follows:

| Signal Name | Direction | Width | Description |
|---------|---------|------|-------------------------------------------------|
| 'ring' | Input | 1 | Indicates when the device should be ringing or vibrating |
| 'vibrate_mode' | Input | 1 | Selects vibrate mode; when high, the device vibrates instead of ringing |
| 'ringer' | Output | 1 | Controls the device's ringer; high to ring, low otherwise |
| 'motor' | Output | 1 | Controls the device's motor; high to vibrate, low otherwise |

### demonstration2
Please write a description about the code snippet and a problem such that the given code snippet is exactly the answer to this problem.
[The Code Snippet]
```verilog
module top_module(
        input clk,
        input resetn,
        input [1:0] byteena,
        input [15:0] d,
        output reg [15:0] q);
    always @(posedge clk) begin
            if (!resetn)
                    q <= 0;
            else begin
                    if (byteena[0])
                            q[7:0] <= d[7:0];
                    if (byteena[1])
                            q[15:8] <= d[15:8];
            end
    end
endmodule
```
[Description]
This Verilog module is a top-level module which is used to write data to a 16-bit register. It has five inputs and one output. The inputs are a clock signal (clk), a reset signal (resetn), a 2-bit byte enable signal (byteena), and a 16-bit data signal (d). The output is a 16-bit register (q). The module is triggered by a rising edge of the clock signal. When the reset signal is low, the register is reset to 0. Otherwise, the data signal is written to the register based on the byte enable signal. If the first bit of the byte enable signal is high, the lower 8 bits of the data signal are written to the lower 8 bits of the register. If the second bit of the byte enable signal is high, the upper 8 bits of the data signal are written to the upper 8 bits of the register.
[Problem] Create 16 D flip-flops. It's sometimes useful to only modify parts of a group of flip-flops. The byte-enable inputs control whether each byte of the 16 registers should be written to on that cycle. byteena[1] controls the upper byte d[15:8], while byteena[0] controls the lower byte d[7:0]. resetn is a synchronous, active-low reset. All DFFs should be triggered by the positive edge of clk. This Verilog module, named 'top_module', has the interface designed as follows:

| Signal Name | Direction | Width | Description |
|---------|---------|------|------------------------|
| 'clk' | Input | 1 | Clock signal |
| 'resetn' | Input | 1 | Active low reset signal |
| 'byteena' | Input | 2 | Byte enable signal |
| 'd' | Input | 16 | Data input signal |
| 'q' | Output | 16 | 16-bit register output |

(continued)
### demonstration3
Please write a description about the code snippet and a problem such that the given code snippet is exactly the answer to this problem.
[The Code Snippet]
```verilog
module top_module(
        input clk,
        input reset,
        output reg [3:0] q);

        always @(posedge clk)
                if (reset || q == 10)
                        q <= 1;
                else
                        q <= q+1;

endmodule
```
[Description]
This top Verilog module is a simple counter that increments its output q by one every clock cycle. It has 3 inputs, a clock (clk), a reset signal (reset), and an output register (q). The output register is a 4-bit register, meaning it can store values from 0 to 15. The module is triggered on the rising edge of the clock signal. When the reset signal is active, the output register is set to 1. If the reset signal is not active, the output register is incremented by one. When the output register reaches 10 (1010 in binary), it is reset to 1. This process is repeated every clock cycle.
[Problem]
Make a decade counter that counts 1 through 10, inclusive. The reset input is active high synchronous, and should reset the counter to 1. This Verilog module, named 'top_module', has the interface designed as follows:
| Signal Name | Direction | Width | Description |
|————-|————|———|————————————————————————|
| 'clk' | Input | 1 | Clock signal that triggers the counter on its rising edge |
| 'reset' | Input | 1 | Active-high synchronous reset signal to initialize the counter|
| 'q' | Output | 4 | 4-bit register output representing the current count (1-10) |

### demonstration4 Please write a description about the code snippet and a problem such that the given code snippet is exactly the answer to this problem.
[The Code Snippet]
```verilog
module top_module(
        input clk,
        input reset,
        output reg [4:0] q);

        logic [4:0] q_next;
        always @(q) begin
                q_next = q[4:1];
                q_next[4] = q[0];
                q_next[2] ^= q[0];
        end

        always @(posedge clk) begin
                if (reset)
                        q <= 5'h1;
                else
                        q <= q_next;
        end
endmodule
```
[Description]
The top module has 3 inputs and 1 output, where the inputs are clk, reset, and output is q. The module has 2 always blocks to define the state transition of q and the logic description. The state transitions are defined in the first always block, which is triggered when q changes. In the first always block, q_next is assigned with q[4:1], which is the value of q except the LSB bit. Then, q_next[4] is assigned with q[0], which is the LSB bit. Lastly, q_next[2] is xored with q[0]. The second always block is triggered at positive edge of clk. If reset is active, q is assigned with 5'h1, which is the reset value. If reset is inactive, q is assigned with q_next, which is the state transition. The port connections of instantiated modules are shown above. The module takes clk, reset, and q as input. q is a 5 bit output, which is assigned with 5'h1 at reset and q_next at positive edge of clk.
[Problem]
A linear feedback shift register is a shift register usually with a few XOR gates to produce the next state of the shift register. A Galois LFSR is one particular arrangement where bit positions with a "tap" are XORed with the output bit to produce its next value, while bit positions without a tap shift. If the taps positions are carefully chosen, the LFSR can be made to be "maximum-length". A maximum-length LFSR of n bits cycles through 2**n-1 states before repeating (the all-zero state is never reached). Build a 5-bit maximal-length Galois LFSR with taps at bit positions 5 and 3. The active-high synchronous reset should reset the LFSR output to 1. This Verilog module, named 'top_module', has the interface designed as follows:
| Signal Name | Direction | Width | Description |
|————-|————|———|————————————————————————|
| 'clk' | Input | 1 | Clock signal that triggers state transitions on rising edges. |
| 'reset' | Input | 1 | Active-high synchronous reset signal to initialize the LFSR. |
| 'q' | Output | 5 | Current state of the LFSR, representing a 5-bit value ('00001' to '11111'). |

### demonstration5
Please write a description about the code snippet and a problem such that the given code snippet is exactly the answer to this problem.
[The Code Snippet]
```verilog
module top_module (
        input [99:0] in,
        output [98:0] out_both,
        output [99:1] out_any,
        output [99:0] out_different
);

        assign out_both = in & in[99:1];
        assign out_any = in | in[99:1];
        assign out_different = in^{in[0], in[99:1]};
endmodule
```
[Description]
This Verilog module is used to compare two input signals and generate three output signals. The first input signal is a 100-bit wide vector, and the second input signal is the same vector shifted by one bit. The module has three output signals, out_both, out_any, and out_different. The out_both signal is generated by performing a bit-wise AND operation between the two input signals. This will result in a 99-bit wide vector, where each bit is 1 only if both the corresponding bits of the two input signals are 1. The out_any signal is generated by performing a bit-wise OR operation between the two input signals. This will result in a 100-bit wide vector, where each bit is 1 if either of the corresponding bits of the two input signals is 1. The out_different signal is generated by performing a bit-wise XOR operation between the two input signals. This will result in a 100-bit wide vector, where each bit is 1 only if the corresponding bits of the two input signals are different. The first bit of the out_different signal is generated by performing a bit-wise XOR operation between the first bit of the first input signal and the last bit of the second input signal.
[Problem]
You are given a 100-bit input vector in[99:0]. We want to know some relationships between each bit and its neighbour:
// (1) out_both: Each bit of this output vector should indicate whether both the corresponding input bit and its neighbour to the left are '1'. For example, out_both[98] should indicate if in[98] and in[99] are both 1. Since in[99] has no neighbour to the left, the answer is obvious so we don't need to know out_both[99].
// (2) out_any: Each bit of this output vector should indicate whether any of the corresponding input bit and its neighbour to the right are '1'. For example, out_any[2] should indicate if either in[2] or in[1] are 1. Since in[0] has no neighbour to the right, the answer is obvious so we don't need to know out_any[0].
// (3) out_different: Each bit of this output vector should indicate whether the corresponding input bit is different from its neighbour to the left. For example, out_different[98] should indicate if in[98] is different from in[99]. For this part, treat the vector as wrapping around, so in[99]'s neighbour to the left is in[0].
This Verilog module, named 'top_module', has the interface designed as follows:
| Signal Name | Direction | Width | Description |
|—————|—————|————|———————————————————————|
| 'in' | Input | 100 | 100-bit input vector for analyzing bit relationships |
| 'out_both' | Output | 99 | Each bit indicates if both the corresponding input bit and its left neighbor are '1' |
| 'out_any' | Output | 99 | Each bit indicates if either the corresponding input bit or its right neighbor is '1' |
| 'out_different' | Output | 100 | Each bit indicates if the corresponding input bit is different from its left neighbor, circularly |

### Instruction
Please write a description about the code snippet and a problem such that the given code snippet is exactly the answer to this problem.
[The Code Snippet]
```verilog
{The Given Code Snippet}
```
### Response

Our prompt begins by presenting five distinct demonstrations. Each demonstration first provides a description of a code snippet, followed by the generation of a corresponding problem. We then prompt the model (DeepSeek-V3) to generate a problem similarly based on the given code snippet colored in red. This process mirrors the multi-level summarization mechanism in CodeV [48].

We also show the system prompt we use during training (both SFT and RL) and testing (on benchmarks) as below.

---
**System Prompt for Training and Testing**

You are a helpful assistant. The assistant first thinks about the reasoning process in the mind and then provides the user with the answer. The reasoning process and answer are enclosed within <think> </think> and<answer> </answer> tags, respectively, i.e., <think> reasoning process here </think><answer> answer here </answer>. Now the user asks you to write verilog code. After thinking, when you finally reach a conclusion, enclose the final verilog code in ```verilog ```within <answer> </answer> tags. i.e., <answer> ```verilog\n module top_module(in, out, ...); ... ```</answer>.

---

# F   Broader Impacts

Through distillation from DeepSeek-R1 and reinforcement learning, CodeV-R1-7B even outperforms DeepSeek-R1-671B on RTLLM v1.1 and RTLLM v2, while outperforming previous Verilog-domain state-of-the-art models (typically 7 15B) by 12∼21 % on RTLLM v1.1 and v2. Through these results, our work demonstrates the promising potential of reinforcement learning for improving circuit design.

However, analogous to other code generation models, CodeV-R1-7B may produce code that misaligns with user intentions or even be misused for unintended purposes. As comprehensively analyzed in broader impact studies [1], such risks include but are not limited to:

1. Functional misalignment: Generated code might superficially satisfy requirements but fail to execute as intended, particularly in safety-critical circuit designs.

2. Security vulnerabilities: The model could inadvertently generate insecure code (e.g., flawed logic or backdoors), which poses risks in hardware deployment.

3. Misuse in malicious contexts: Lower barriers to code generation may facilitate the creation of obfuscated or harmful designs, especially as model capabilities scale.

Given the potentially severe consequences of such issues in hardware systems, we strongly recommend that users:

1. Conduct rigorous functional verification and security audits for all generated code.

2. Implement access controls and usage monitoring to mitigate abuse risks.

3. Adopt a principle of "human-in-the-loop" oversight, particularly for high-stakes applications.

# G   Limitations and Future Work

This work has several limitations, and we primarily discuss two key aspects that also define our future direction: (1) The automated testbench generation framework can only improve the semantic consistency between code and NL in the probabilistic sense. The synthetic dataset generated by our method both for SFT and RL may still contain a small amount of low-quality data, which could potentially impact the model's performance. (2) Collecting data with reasoning processes for SFT requires a general reasoning model (e.g., DeepSeek-R1), which inherently depends on the teacher model's reasoning capabilities. This dependency poses greater challenges in specialized domains where the teacher model's performance is suboptimal, as its limitations in such contexts may directly impact the quality of the collected data. Besides, this process might be financially costly.

Additionally, from an application perspective, it is promising to focus on exploring the potential of reasoning LLMs to tackle more complex hardware development tasks beyond RTL code generation in the future, such as PPA performance optimization and analog circuit synthesis.

