# OpenReview forum: "QiMeng-CodeV-R1: Reasoning-Enhanced Verilog Generation"
_NeurIPS.cc/2025/Conference — NeurIPS 2025 poster_

### Official Review · Reviewer_oSjb · 2025-07-02

**Clarity:** 3
**Significance:** 2
**Originality:** 2
**Rating:** 4
**Confidence:** 4

**Summary:**

This paper introduces ReasoningRTL, a Reinforcement Learning with Verifiable Reward (RLVR) framework designed for generating Register-Transfer Level (RTL) hardware descriptions. The proposed framework comprises three main components:
- A rule-based testbench generator that produces test cases to verify the correctness of RTL code generated by LLMs;
- A natural language description generator that converts RTL code snippets into natural language descriptions to serve as inputs for model training;
- A training framework that integrates cold-start SFT and adaptive DAPO to enhance the model's RTL generation capabilities.

The framework is used to train a 7B-parameter LLM, which is then evaluated on multiple RTL generation benchmarks.

**Questions:**

N/A

**Ethical Concerns:**

["NO or VERY MINOR ethics concerns only"]

**Final Justification:**

The authors have addressed my earlier concerns regarding the experimental results. However, I remain unconvinced about the relevance of Definition 2.1 and Theorem 2.1 to the rest of the paper. In particular, the assumption of determinism in generating programs and natural language descriptions seems misaligned with the nature of RTL design, where multiple valid representations and descriptions naturally exist. I have therefore adjusted my score to 4.

**Limitations:**

yes

**Quality:**

3

**Strengths And Weaknesses:**

trengths:
+ The paper is generally well-written and well-structured.
+ The natural language description generator addresses a key bottleneck in RTL generation: the scarcity of high-quality, task-specific training data.
+ The approach is relatively cost-effective, requiring only 2656 A100-GPU hours for training, which is competitive for LLM development.

Weaknesses:
- The evaluation results require further clarification. The proposed model shows a notable improvement over existing methods on the RTLLM dataset, yet achieves similar or even worse performance on the VerilogEval benchmark. This raises questions about dataset-specific characteristics and should be more thoroughly discussed. What makes RTLLM particularly amenable to improvement under this framework?
- Definition 2.1 and Theorem 2.1 are confusing. Given the inherently probabilistic nature of LLMs, the conditions in Definition 2.1 are unlikely to hold in practice, which undermines the applicability and significance of Theorem 2.1. Moreover, they appear to have limited relevance to the rest of the paper.
- The natural language description generator component would benefit from a deeper analysis. While the appendix includes some illustrative examples, they are limited to short RTL code snippets. In practice, generating natural language descriptions for complex RTL modules, especially those involving non-trivial sequential logic, is a challenging task. Without access to simulation traces, inferring sequential behavior from code alone is difficult even for human experts. This raises concerns about the generator's ability to generalize to realistic and more complex RTL designs.

---

> ### Author Rebuttal · Authors · 2025-07-31
>
> We sincerely thank you for your comprehensive feedback, particularly for your appreciation of our writing, training data generation process, and training framework. For your concerns on our work's cross-benchmark performance variations, theoretical foundations, and NL generation for complex modules, we would like to respond as follows:
>
> - **Q1:** The evaluation results require further clarification. The proposed model shows a notable improvement over existing methods on the RTLLM dataset, yet achieves similar or even worse performance on the VerilogEval benchmark. This raises questions about dataset-specific characteristics and should be more thoroughly discussed. What makes RTLLM particularly amenable to improvement under this framework?
>
> Thank you for your focus on dataset-specific characteristics. Given that RTLLM's performance gains stem mainly from reinforcement learning, we analyzed distribution differences between our RL dataset and the two benchmarks. We run the instructer-embedding model [1] for the golden code in our RL dataset, RTLLM (v2), and VerilogEval (v2 spec-to-RTL), then generated a t-SNE distribution plot [2]. This plot revealed our RL dataset aligns closely with RTLLM but diverges from VerilogEval (to be added in revision). Below we also show cosine similarity and Euclidean distance metrics for the **embedding centroids** between our RL dataset and the benchmarks:
>
> |                    | RTLLM v2 | VerilogEval v2 |
> | ------------------ | -------- | -------------- |
> | Euclidean Distance | 0.1150   | 0.2903         |
> | Cosine Similarity  | 0.9926   | 0.9547         |
>
> Our RL dataset’s embedding centroid shows significantly smaller Euclidean distance and closer-to-1 (maximum value) cosine similarity with RTLLM than with VerilogEval. Importantly, this reflects only **embedding centroid** relationships, not data homogenization or overfitting to RTLLM.
>
> Additionally, we conducted a detailed case study across both datasets and identified one reason is VerilogEval’s heavier use of table/graph-based problems, where our model underperforms significantly. Below is a comparison of our model's accuracy against Deepseek-r1 on these problem types, including their ratios within benchmarks:
>
> |                           | Tables/Graphs (Ours) | Tables/Graphs (DS-R1) |
> | ------------------------- | -------------------- | --------------------- |
> | VerilogEval v2 (Ratio)    | 29.49%               | 29.49%                |
> | VerilogEval v2 (Accuracy) | 55.43%               | 78.04%                |
> | RTLLM v2 (Ratio)          | 0%                   | 0%                    |
> | RTLLM v2 (Accuracy)       | N/A                  | N/A                   |
>
> This points to a key improvement direction: incorporating more table/graph-specific instruction-response pairs (e.g., KMap, FSM, waveform data as in CraftRTL [3]) into our training dataset.
>
> Finally, we should mention that RTLLM is more practical than VerilogEval since VerilogEval clearly states "we do not evaluate the performance of downstream circuit implementations". The similarity between our dataset and RTLLM shows the practicality of our dataset. In addition, we have gone through rigorous data decontamination against these benchmarks by Rouge-L score during data collection (in section 2.2).
>
> [1] Su, H., Shi, W., Kasai, J., Wang, Y., Hu, Y., Ostendorf, M., ... & Yu, T. (2022). One embedder, any task: Instruction-finetuned text embeddings. arXiv preprint arXiv:2212.09741.
>
> [2] Maaten, L. V. D., & Hinton, G. (2008). Visualizing data using t-SNE. Journal of machine learning research, 9(Nov), 2579-2605.
>
> [3] Liu, M., Tsai, Y. D., Zhou, W., & Ren, H. (2024). Craftrtl: High-quality synthetic data generation for verilog code models with correct-by-construction non-textual representations and targeted code repair. arXiv preprint arXiv:2409.12993.
>
>
>
> - **Q2:** Definition 2.1 and Theorem 2.1 are confusing. Given the inherently probabilistic nature of LLMs, the conditions in Definition 2.1 are unlikely to hold in practice, which undermines the applicability and significance of Theorem 2.1. Moreover, they appear to have limited relevance to the rest of the paper.
>
> We sincerely thank you for your thoughtful critique regarding our theoretical framework.
>
> As for the stringency of Theorem 2.1's conditions, although LLMs behave probabilistically in text, they can be relatively deterministic when implementing the functionality. For instance, given the NL prompt "Generate a Python implementation of the quicksort algorithm," the specific code text produced might vary slightly (e.g., in variable naming or comment placement), but the core functionality $f$ (i.e., a correctly working quicksort algorithm in Python) remains highly consistent. Notably, the stronger the model's capability and the more specific the NL description, the more deterministic this $L \rightarrow F$ (NL->code) mapping becomes. This logic also applies in the reverse direction, where generating an NL description ($l$) from a given code ($f$) can exhibit similar determinism under sufficient model competence. We will clarify this in our revised paper.
>
> Definition 2.1 and Theorem 2.1 are relevant to the rest of the paper based on the above insights. **They provide a theoretical foundation for understanding why the round-trip data synthesis process can enhance data quality**: as model capacity approaches the assumptions outlined in Definition 2.1 (e.g., more reliable bidirectional mappings), the outputs of the synthesis process align more closely with the guarantees in Theorem 2.1, resulting in higher-quality data. Specifically, the deterministic equivalence of generated code and source code will guarantee the correctness of both problem summarization and code generation. Our experimental results further validate the reasonableness of this theoretical framework.
>
> In conclusion, these definitions and theorems offer critical insights from an information-theoretic perspective, elucidating the underlying mechanisms through which round-trip synthesis reduces noise and strengthens the consistency between input and output spaces—ultimately justifying its effectiveness in improving data quality.
>
>
>
> - **Q3:** The natural language description generator component would benefit from a deeper analysis. While the appendix includes some illustrative examples, they are limited to short RTL code snippets. In practice, generating natural language descriptions for complex RTL modules, especially those involving non-trivial sequential logic, is a challenging task. Without access to simulation traces, inferring sequential behavior from code alone is difficult even for human experts. This raises concerns about the generator's ability to generalize to realistic and more complex RTL designs.
>
> We sincerely thank you for your insightful perspective on natural language generation for complex RTL. We would like to engage with this perspective through the following technical considerations:
>
> Our dataset contains significantly more complex code than the limited examples included in the appendix. This underscores the capability of our natural language description generator: it can effectively handle such complex code, as evidenced by the high-performance model we trained using the dataset—composed entirely of (problem, code) pairs generated by this generator. Notably, our model can solve asyn_fifo (147 lines of golden code) and float_multi (165 lines) in RTLLM v2. Below is a table comparing the average code length (in tokens) in our dataset with that of the benchmarks (RTLLM v2 and VerilogEval v2 - Spec to RTL).
>
> |          | SFT dataset | RTLLM v2 | VerilogEval v2 |
> | -------- | ----------- | -------- | -------------- |
> | RTL code | 478.50      | 403.80   | 122.31         |
> | Prompt   | 529.81      | 348.82   | 219.94         |
>
> From the table, the average code length in our dataset is longer than that in RTLLM v2 and VerilogEval v2. Besides, the longest code in our SFT dataset is **over 3000 tokens**, and those can pass the round-trip equivalence checking in our RL dataset include relatively complex ones like **single round of SHA-256 calculation**.
>
> We also acknowledge that, although we have trained a high-performance model using our workflow involving this natural language description generator, we have room to further improve this process. For example, we can add mechanisms like multi-turn refinement for code summarization with RTL simulator, code parser or other tools.
>
> Thank you again for your valuable feedback. Should our responses adequately address your concerns, we would be grateful for your consideration in recognizing these improvements in your final assessment. If any questions remain, we welcome further discussion.

---

> > ### Comment · Area_Chair_PfmX · 2025-08-04
> >
> > Dear Reviewer oSjb,
> >
> > Thank you for your time and effort in reviewing the submissions and providing valuable feedback to the authors.
> >
> > If you haven't already done so, we kindly remind you to review the authors' rebuttals and acknowledge them by clicking the "Mandatory Acknowledgement" button at your earliest convenience. This step ensures efficient communication and helps finalize the process smoothly.
> >
> > We sincerely appreciate your dedication and collaboration.
> >
> > Best,
> >
> > Your AC

---

> > ### Comment · Reviewer_oSjb · 2025-08-08
> >
> > The authors have addressed my earlier concerns regarding the experimental results. However, I remain unconvinced about the relevance of Definition 2.1 and Theorem 2.1 to the rest of the paper. In particular, the assumption of determinism in generating programs and natural language descriptions seems misaligned with the nature of RTL design, where multiple valid representations and descriptions naturally exist. I have therefore adjusted my score to 4.

---

> > > ### Author Response · Authors · 2025-08-09
> > >
> > > Thank you for acknowledging that our supplementary experiments have addressed your earlier concerns. We want to elaborate on the relevance of Definition 2.1 and Theorem 2.1 in the following.
> > >
> > > In Definition 2.1 and Theorem 2.1, we actually define the space $F$ as the space of **all code functions instead of all code snippets**. That is to say, all valid representations of the same function (e.g., a full adder) represent the same point in space $F$ (the same to $L$).  The determinism means that the **function mapping** from code to description ($f \rightarrow l$) and from description to code ($l \rightarrow f$) is deterministic (just as we have explained, given the NL prompt "Generate a Python implementation of the quicksort algorithm," the specific code text produced might vary slightly, but the core functionality remains highly consistent).
> > >
> > > Intuitively, this theorem shows that ideally when the generated code ($Y' \in F$) and source code ($Y \in F$) are equivalent, our problem summarization process (code $Y \in F$ to NL $X \in L$) and the code generation process (NL $X$ to code $Y'$) should both be correct. That is to say, the problem correctness in our RL dataset (equivalence between code $Y$ and NL $X$) can be supported by the equivalence of generated code ($Y'$) and source code ($Y$) in our round-trip synthesis.
> > >
> > > Thank you again for your reviewing and we will clarify this in the revised paper.

---

> ### Author Response · Authors · 2025-08-06
>
> Dear reviewer, we sincerely thank you for taking the time to review our paper. As the discussion phase is coming to an end, we note that you have read our response but have not raised any further discussion. We believe that our response addresses your main concerns. We would greatly appreciate it if you could review our response and participate in the discussion by indicating whether your concerns have been addressed. This will help ensure a fair and comprehensive evaluation of our work. Thank you again for your valuable contribution.

---

> ### Author Response · Authors · 2025-08-08
>
> Dear Reviewer,
>
> We hope this message finds you well. As the discussion period is nearing its end (extended to Aug 8, 11.59 pm AoE), we want to ensure that we have addressed all of your concerns satisfactorily. If there are any additional suggestions or feedback you'd like us to consider, please do not hesitate to let us know. Your insights are invaluable to us, and we're eager to address any remaining concerns to improve our work.
>
> Thank you for your time and effort in reviewing our paper.

---

### Official Review · Reviewer_NGaP · 2025-07-02

**Clarity:** 3
**Significance:** 3
**Originality:** 2
**Rating:** 5
**Confidence:** 4

**Summary:**

The paper introduces ReasoningRTL (R2TL), , a framework to train RTL generation LLM via reinforcement learning with verifiable reward. The framework consists of 3 main contributions: a rule-based testbench generator that performs robust equivalence checking against
golden references, a round-trip data synthesis method that creates a high-quality dataset via validating equivalence between original and regenerated code, a two-stage training pipeline for supervised distillation and an efficient RL algorithm (adaptive DAPO). Evaluation shows it surpasses prior state-of-the-art by 12∼21%.

**Questions:**

- How could the R^2TL be potentially combined with agentic methods? Will it still outperform large size model? e.g, MAGE: A Multi-Agent Engine for Automated RTL Code Generation (DAC 2025), which demonstrates  95% pass rate for VerilogEval dataset.
- What is the complexity of the RTL code (#tokens) obtained from GitHub? Any case with a very long context length that fails with the 16k context length?
- How does the rule-based test bench generation can be generalized to larger-scale or complex modules/ hierarchical modules?
- What could be some potential solutions to further reduce the 2.7% wrong flags?
- How does the method can be potentially generalized to other HDLs like VHDL, System Verilog?

**Ethical Concerns:**

["NO or VERY MINOR ethics concerns only"]

**Final Justification:**

The paper addressed an important problem of generate high-accuracy RTL code using 7B models, by integrating several key techniques (RLVR, rule-based test generation, data synthesis) in a novel way. The results are promising. The rebuttal addressed my concerns.

**Limitations:**

See weakness

**Quality:**

3

**Strengths And Weaknesses:**

Strengths:
- Applying RLVR to RTL generation domain is a new yet important scope.
- The rule-based test-generation is novel and achieves 35.5 % fewer false negatives and 62.5 % more injected errors detection in fuzzing tests.
- The round trip data synthesis method is novel and ensures a high-quality training data.
- The adaptive DAPO leads to 1.25x speedup for RL
- The framework shows 12-21% improvement over prior SOTA with a 7B model, even better than 671B DeepSeek-R1 in some testing datasets.

Weaknesses:
- The rule-based test bench generation may not be good enough for complex or real-world Verilog design, requiring manual maintaining with more expert heuristics.
- While the method outperforms SOTA on RTLLM (v1.1 and v2), it underperforms on VerilogEval, I believe the RTLLM has fewer tests and is relatively easier than VerilogEval.
- 3.1k data is small (while demonstrated high-quality), which may affect the generalization and robustness of RL training, even induce overfitting to the dataset distribution. This is not well verified and discussed in the paper.

---

> ### Author Rebuttal · Authors · 2025-07-31
>
> Thank you for your valuable feedback. We appreciate your recognition of our contributions, including our method and model performance. Your insights on the limitations, including rule-based testbenches’ scalability, VerilogEval performance, training data size and complexity concerns, and agent performance, are crucial and will guide our improvements. We will address your concerns as follows:
>
> - **W1&Q3:** The rule-based testbench generation may require manual maintaining with more expert heuristics for complex or real-world Verilog design. How to generalize it to larger-scale or complex/hierarchical modules?
>
> Thank you for pinpointing the challenges of rule-based approaches in complex Verilog verification. Our testbench, built on manual rules, runs fully automatically and works directly for **complex multi-module designs like SHA256 or FFT**, validating DUT-golden code equivalence via consistent I/O checks.
>
> Limitations of our framework include unguaranteed coverage with random I/O, no support for multiple clocks, and certain syntax issues (e.g., macro definitions, Verilog-SystemVerilog reserved word differences). A future direction is to added LLM-generated directed cases to target corner cases missed by random stimuli.
>
> - **W2:** While the method outperforms SOTA on RTLLM, it underperforms on VerilogEval, I believe the RTLLM has fewer tests and is relatively easier than VerilogEval.
>
> We sincerely thank you for highlighting the performance differences across benchmarks. We emphasize that RTLLM is **harder and more practical** than VerilogEval. Moreover, our model has also achieved **SOTA results among 7B models** on VerilogEval v1-Human (surpassing the previous SOTA model CraftRTL-DS-6.7B by 4.5%) and VerilogEval v2.
>
> In terms of practicality, VerilogEval [1] explicitly states that "we do not evaluate the performance of downstream circuit implementations, a gap that is addressed by the work presented in RTLLM". Therefore, RTLLM is more aligned with real-world applications. The test coverage on RTLLM is strong, with 98.5% average line coverage for v1.1 and 96.0% for v2, which is satisfactory.
>
> Regarding difficulty, results in Tables 1&2 show most models (except Qwen2.5-Coder and RTLCoder-Mistral-7B) perform better on VerilogEval than RTLLM, including our distillation model R2TL-7B-Distill. RTLLM is also harder by code length -- The table below provides the average number of lines and tokens (comments and blank lines removed) for VerilogEval and RTLLM:
>
> | Dataset           | Avg Lines | Avg Tokens |
> | ----------------- | --------- | ---------- |
> | VE1 Machine       | 13.89     | 93.85      |
> | VE1 Human         | 15.82     | 117.35     |
> | VE2 Code Complete | 16.10     | 121.84     |
> | VE2 Spec to RTL   | 16.12     | 122.31     |
> | RTLLM 1.1         | 56.52     | 470.48     |
> | RTLLM 2.0         | 46.30     | 403.80     |
>
> Meanwhile, our dataset's distribution is more consistent with RTLLM than VerilogEval. Details can be found in the answer for Q1 for reviewer oSjb.
>
> [1] Liu, M. et al. Verilogeval: Evaluating large language models for verilog code generation.
>
> [2] Su, H. et al. One embedder, any task: Instruction-finetuned text embeddings.
>
> - **W3:** 3.1k data is small (although high-quality), which may affect the generalization and robustness of RL training, even induce overfitting to the dataset distribution. This is not well verified and discussed in the paper.
>
> We appreciate your note on the dataset scale. Our model shows strong generalization on unseen benchmark problems, as evidenced by the performance gain from R2TL-7B-Distill (without RL) to R2TL-7B (with RL) on VerilogEval and RTLLM (Tables 1&2). Note we have rigorously de-contaminated our training data against benchmarks to avoid overfitting as shown in Section 2.2.
>
> Moreover, 3.1k is typical for RL in this context. LLM-based RL prioritizes data quality over quantity: Qwen3 [1] utilized 3995 query-verifier pairs during their reasoning RL phase. LIMR [2] filtered 8523 RL data down to 1389 and achieved superior results. Logic-RL [3] enhanced models' logical reasoning capabilities using a mere 900 instances. All these works demonstrate that 3.1k data is sufficient for our RL setup. We will clarify this in the revised paper.
>
> [1] Yang, A. et al. Qwen3 technical report.
>
> [2] Li, X. et al. Limr: Less is more for rl scaling.
>
> [3] Xie, T. et al. Logic-rl: Unleashing llm reasoning with rule-based reinforcement learning.
>
> - **Q1:** How could R2TL be potentially combined with agentic methods? Will it still outperform large size model? e.g, MAGE: A Multi-Agent Engine for Automated RTL Code Generation, which demonstrates 95% pass rate for VerilogEval dataset.
>
> Thank you for your forward-looking inquiry regarding agentic integration. We note two fundamental distinctions between MAGE and R2TL: First, the MAGE agent uses the golden testbench while R2TL does not. Second, the performance of MAGE strongly depends on prompt engineering and is model-dependent (see table below), which does not align well with the model capacity.
>
> Given these divergences and computational constraints (MAGE's multi-agent sequential calls limiting parallelization efficiency, as well as the time limitation of rebuttal period), we designed a targeted experiment, using R2TL to replace the RTL generation agent (which best aligns R2TL's ability among the four agents) in MAGE, keeping other agents based on Claude. The test results are shown in the table below.
>
> | Model | Pass@1  |
> | - | - |
> | Deepseek-R1-0528 | 80.77 % |
> | Deepseek-V3-0324 | 88.46 % |
> | GPT-4o-240806 | 76.28 % |
> | Claude-3.5-Sonnet | 95.51 % |
> | R2TL (with claude) | 94.87 % |
>
> These results demonstrate the potential of R2TL to be applied in the agent workflow, which implies R2TL could develop end-to-end agent capabilities in the future.
>
> - **Q2:** What is the complexity of the RTL code (#tokens) obtained from GitHub? Any case with a very long context length that fails with the 16k context length?
>
> Thank you for raising this point. Below are some statistics of our 87k distillation dataset:
> |  | Mean   | Max   | % over 16k tokens |
> | - | - | - | - |
> | RTL code | 537.5  | 3375  | 0.0% |
> | Response (DS-R1 CoT) | 6301.5 | 39927 | 0.5% |
> | Prompt+Response | 6831.3 | 40319 | 0.8% |
>
> Distillation considers (prompt + response) length: around 0.8% of the data exceed the 16k context length and truncated. Meanwhile, the longest RTL code in our dataset is 3375 tokens. Some examples of long code include single round of SHA-256 calculation, CORDIC arctan() implementation, BCD-format 12-hour clock with AM/PM, etc.
>
> - **Q4:** What could be some potential solutions to further reduce the 2.7% wrong flags?
>
> Thank you for your concern about this. The 2.7% mismatch in golden-vs-golden comparisons mainly stems from corner syntax errors, like inconsistent variable names, generated by the automated testbench, while the runtime inequivalences only count for **0.3%**. In contrast, the 7.6% inaccuracy for LLM-generated testbenches excludes such test-phase errors. Using a consistent standard, our method’s false negative rate is less than 1/20 of the LLM approach, showing superior stability.
>
> For the remaining inequivalences, around half involve random numbers, which fall outside our testbench's scope. For the remaining half, removing the `-g2012` compile option works for some cases, and **switching from Icarus Verilog to VCS** can work for most cases. (However, VCS requires license so it is not as convenient as Icarus Verilog.)
>
> - **Q5:** How does the method can be potentially generalized to other HDLs like VHDL, System Verilog?
>
> We appreciate your question about HDL generalization. Our method is adaptable to other HDLs: we already support SystemVerilog (for VerilogEval) and Verilog (for RTLLM). To extend to other HDLs like VHDL, our five-stage framework (Figure 1) can be adjusted for specific languages:
>
> In pre-Stage 1 data preparation, we would collect language-specific code snippets (e.g., VHDL modules) from relevant repositories, similar to how we gathered Verilog from GitHub.
>
> For Stages 3-5, our automated testbench framework would be extended to support the target HDL (mainly syntax adaptation), enabling proper verification during difficulty filtering (Stages 3,5) and equivalence checking (Stage 4).
>
> With such adjustments, R2TL can generalize to more HDLs beyond Verilog and SystemVerilog.
>
> Thank you again for this valuable discussion. Please feel free to ask if you still have any questions.

---

> > ### Comment · Area_Chair_PfmX · 2025-08-04
> >
> > Dear Reviewer NGaP,
> >
> > Thank you for your time and effort in reviewing the submissions and providing valuable feedback to the authors.
> >
> > If you haven't already done so, we kindly remind you to review the authors' rebuttals and acknowledge them by clicking the "Mandatory Acknowledgement" button at your earliest convenience. This step ensures efficient communication and helps finalize the process smoothly.
> >
> > We sincerely appreciate your dedication and collaboration.
> >
> > Best,
> >
> > Your AC

---

> > ### Comment · Reviewer_NGaP · 2025-08-06
> >
> > Thank you for your response addressing my concerns. I am happy to see that R2TL could be potentially integrated with agent methods.
> >
> > When you state that 'RTLLM is harder and more practical than VerilogEval,' are you basing this judgment on the average number of tokens in the golden RTL code? Which tokenizer are you using? Why removing comments -- considering comments could be the valuable context in generation.

---

> > > ### Author Response · Authors · 2025-08-06
> > >
> > > Thank you for your feedback.
> > >
> > > For difficulty comparison between benchmarks, we are based on the average number of tokens in the golden RTL code. For practicality, we are based on the statement in VerilogEval [1]: "We do not evaluate the performance of downstream circuit implementations, a gap that is addressed by the work presented in RTLLM".
> > >
> > > We are using the Qwen2.5-Coder-7B-Instruct (which is the base model of R2TL). We remove comments to compare the pure code complexity. Below, we also provide the statistics before removing comments and blank lines.
> > >
> > > | Dataset           | Avg Lines (Things Removed) | Avg Tokens (Things Removed) | Avg Lines (Things Kept) | Avg Tokens (Things Kept) |
> > > | ----------------- | -------------------------- | --------------------------- | ----------------------- | ------------------------ |
> > > | VE1 Machine       | 13.89                      | 93.85                       | 16.36                   | 97.19                    |
> > > | VE1 Human         | 15.82                      | 117.35                      | 18.60                   | 121.03                   |
> > > | VE2 Code Complete | 16.10                      | 121.84                      | 21.15                   | 125.19                   |
> > > | VE2 Spec to RTL   | 16.12                      | 122.31                      | 21.19                   | 125.77                   |
> > > | RTLLM 1.1         | 56.52                      | 470.48                      | 66.45                   | 486.45                   |
> > > | RTLLM 2.0         | 46.30                      | 403.80                      | 54.72                   | 425.58                   |
> > >
> > > Thank you again for your feedback and for taking the time to review this paper.
> > >
> > > ```
> > > [1] Liu, M. et al. Verilogeval: Evaluating large language models for verilog code generation.
> > > ```

---

> > > > ### Comment · Reviewer_NGaP · 2025-08-06
> > > >
> > > > But in this paper, you did not include any performance (PPA) of downstream circuit implementations, right?
> > > >
> > > > Also, can you give some intuition on why many previous works, including but not limited to:
> > > > [1] RTLCoder: Fully Open-Source and Efficient LLM-Assisted RTL Code Generation Technique
> > > > [2] Towards llm-powered verilog rtl assistant: Self-verification and self-correction
> > > > [3] Leader board: https://iprc-dip.github.io/Chip-Design-LLM-Zoo/
> > > > all showing higher pass@k scores on RTLLM than VerilogEval-Human?
> > > >
> > > > Thank you

---

> > > > > ### Author Response · Authors · 2025-08-06
> > > > >
> > > > > Thank you for your further feedback.
> > > > >
> > > > > We apologize that we initially took "performance" in "We do not evaluate the performance of downstream circuit implementations, a gap addressed by [13]" to mean functional correctness (given its paragraph’s focus), but now agree it refers to PPA performance.
> > > > >
> > > > > For model accuracy on theses two benchmarks, most works typically report both pass@1 and pass@5 on VerilogEval, while only report pass@5 on RTLLM ([1] [2] [3] [4] [15] [17], the data in [3] are consistent with pass@5 for RTLLM v1.1, e.g. RTLCoder, CodeV, AutoVCoder). Below is a table of models evaluated on both VerilogEval v1-Human pass@5 and RTLLM v1.1 pass@5 (including those listed in [1] [2] [3]).
> > > > >
> > > > > | Model | VerilogEval-Human pass@5 (%) | RTLLM v1.1 pass@5(%) | Diff |
> > > > > | - | - | - | - |
> > > > > | CodeGen-Verilog-16B [7] | 43.9 | 24.1 | 19.8 |
> > > > > | CraftRTL-CL-7B [9] | 67.8 | 52.9 | 14.9 |
> > > > > | RTLCoder-DeepSeek-4bit [20] | 50.5 | 37.9 | 12.6 |
> > > > > | CraftRTL-DS-6.7B [9] | 70.0 | 58.8 | 11.2 |
> > > > > | CodeV-CQ-7B [8] | 65.1 | 55.2 | 9.9 |
> > > > > | GPT-3.5 [12] | 45.8 | 37.9 | 7.9 |
> > > > > | QWQ-32B [19] | 78.0 | 70.6 | 7.4 |
> > > > > | CodeV-DS-6.7B [8] | 62.5 | 55.2 | 7.3 |
> > > > > | CraftRTL-SC2-15B [9] | 72.4 | 65.8 | 6.6 |
> > > > > | DeepSeek-R1-Distill-Qwen-7B [10] | 6.3 | 0.0 | 6.3 |
> > > > > | DeepSeek-V3 [11] | 79.7 | 74.2 | 5.5 |
> > > > > | GPT-4o [14] | 71.4 | 65.9 | 5.5 |
> > > > > | DeepSeek-R1 [10] | 87.6 | 82.9 | 4.7 |
> > > > > | AutoVCoder-Codellama-7B [4] | 52.8 | 48.3 | 4.5 |
> > > > > | HAVEN-CodeLlama [15] | 59.0 | 54.7 | 4.3 |
> > > > > | AutoVCoder-CodeQwen-7B [4] | 55.9 | 51.7 | 4.2 |
> > > > > | DeepSeek-R1-Distill-Qwen-32B [10] | 68.1 | 64.3 | 3.8 |
> > > > > | HAVEN-CodeQwen [15] | 64.8 | 62.2 | 2.6 |
> > > > > | AutoVCoder-DeepSeek-6.7B [4] | 53.7 | 51.7 | 2 |
> > > > > | RTLCoder-DeepSeek-6.7B [20] | 50.1 | 48.3 | 1.8 |
> > > > > | RTLCoder-Mistral-4bit [20] | 42.3 | 41.4 | 0.9 |
> > > > > | Starcoder [21] | 26.1 | 27.6 | -1.5 |
> > > > > | HAVEN-DeepSeek [15] | 64.2 | 66.0 | -1.8 |
> > > > > | Qwen2.5-Coder-7B-Instruct [18] | 46.3 | 48.2 | -1.9 |
> > > > > | CodeV-CL-7B [8] | 59.5 | 62.1 | -2.6 |
> > > > > | RTLCoder-Mistral-7B [20] | 45.5 | 48.3 | -2.8 |
> > > > > | Codegen2-16B [6] | 4.1 | 6.9 | -2.8 |
> > > > > | OriGen-DeepSeek-7B [17] | 60.1 | 65.5 | -5.4 |
> > > > > | Claude-3 [5] | 48.3 | 55.2 | -6.9 |
> > > > > | Mistral-7B-v0.1 [16] | 12.6 | 20.7 | -8.1 |
> > > > > | Qwen2.5-Coder-32B-Instruct [18] | 58.1 | 67.7 | -9.6 |
> > > > > | GPT-4 [13] | 55.8 | 65.5 | -9.7 |
> > > > >
> > > > > The table shows more models perform better on VerilogEval, with generally more pronounced advantages. However, some models do perform better on RTLLM, likely due to differing question/answer distributions between the two benchmarks: t-SNE [22] plots of question and golden code embeddings show RTLLM's distributions align better with our RL training set (after rigorous decontamination applied, see response to Reviewer oSjb's Q1 for more details and other metrics); from the case study, 29.49% of the questions in VerilogEval involve tables / diagrams, while all questions in RTLLM are text descriptions. This reflects their distribution differences: RTLLM problems need longer answers, while VerilogEval’s are more diverse (e.g., tables for Karnaugh Maps or Finite State Machines). We suggest the better RTLLM performance may stem from training data distributions more similar to RTLLM's.
> > > > >
> > > > >
> > > > >
> > > > > ```
> > > > > [1] RTLCoder: Fully Open-Source and Efficient LLM-Assisted RTL Code Generation Technique
> > > > > [2] Towards llm-powered verilog rtl assistant: Self-verification and self-correction
> > > > > [3] https://iprc-dip.github.io/Chip-Design-LLM-Zoo/
> > > > > [4] AutoVCoder: A Systematic Framework for Automated Verilog Code Generation using LLMs
> > > > > [5] The Claude 3 Model Family: Opus, Sonnet, Haiku.
> > > > > [6] Codegen2: Lessons for training llms on programming and natural languages
> > > > > [7] Benchmarking large language models for automated verilog rtl code generation
> > > > > [8] CodeV: Empowering LLMs for Verilog Generation through Multi-Level Summarization
> > > > > [9] CraftRTL: High-quality Synthetic Data Generation for Verilog Code Models with Correct-by-Construction Non-Textual Representations and Targeted Code Repair
> > > > > [10] Deepseek-r1: Incentivizing reasoning capability in llms via reinforcement learning
> > > > > [11] Deepseek-v3 technical report
> > > > > [12] Introducing chatgpt
> > > > > [13] GPT-4 technical report
> > > > > [14] GPT-4o system card
> > > > > [15] HAVEN: Hallucination-Mitigated LLM for Verilog Code Generation Aligned with HDL Engineers
> > > > > [16] Mistral 7B
> > > > > [17] OriGen: Enhancing RTL Code Generation with Code-to-Code Augmentation and Self-Reflection
> > > > > [18] Qwen2.5-coder technical report
> > > > > [19] QWQ-32b: Embracing the power of reinforcement learning
> > > > > [20] RTLCoder: Outperforming GPT-3.5 in Design RTL Generation with Our Open-Source Dataset and Lightweight Solution
> > > > > [21] StarCoder: may the source be with you!
> > > > > [22] Visualizing data using t-SNE
> > > > > ```

---

> > > > > > ### Comment · Reviewer_NGaP · 2025-08-06
> > > > > >
> > > > > > Thank you for addressing my concerns. I have raised my score.
> > > > > >
> > > > > > The paper could benefit from including further discussion and analysis on the results for VerilogEval vs. RTLLM (like your response to Reviewer oSjb's Q1).

---

> > > > > > > ### Author Response · Authors · 2025-08-07
> > > > > > >
> > > > > > > Thank you very much for your recognition, and we truly appreciate your valuable feedback and thoughtful suggestions.
> > > > > > >
> > > > > > > Following your advice, we will incorporate the relevant analysis on VerilogEval vs. RTLLM results (including the details from our response to Reviewer oSjb's Q1) into the revised paper to better contextualize the comparative findings and their implications.

---

### Official Review · Reviewer_ukvi · 2025-07-03

**Clarity:** 3
**Significance:** 3
**Originality:** 4
**Rating:** 5
**Confidence:** 4

**Summary:**

This paper introduces R2TL, a tool that uses reasoning to generate RTL. The paper goes into rigorous details on how they generate RTL, assess it's quality, and measure the performance of the code generated quality and the code generate performance.

**Questions:**

+ R2TL versus future reasoners/coder models? Fundamentally, what does R2TL bring to the table beyond SOTA models, and how can it use new models to improve RTL code generation?

**Ethical Concerns:**

["NO or VERY MINOR ethics concerns only"]

**Final Justification:**

Rebuttal helped convince me that my score is justified, and I find this a very good paper and valuable contribution to the community.

**Limitations:**

N/A.

**Quality:**

4

**Strengths And Weaknesses:**

This paper addresses an important problem, namely RTL code generation. RTL code generation is challenging because LLMs typically don't have enough examples to "naturally" produce RTL, plus there are some precision requirements where an LLM cannot make mistakes in (e.g., clk and reset configuration). This makes it a challenging problem, and the paper attempts to address this via reasoning.

I particularly enjoyed the plethora of different ML-based techniques for this system's level problem. This includes the adaptive DAPO for reducing cost, distillation, and focusing on high quality data over too much data. While not particularly novel, the application here to RTL generation is nice, plus the results are decent. Further, the proofs in the appendix, and the data distribution measurements are well received by this reviewer, and hoping more papers employ such rigor in their evaluation.

While not a true weakness: the results aren't mindblowing compared to state-of-the-art methods. So the question then becomes is such a complex approach necessary, or will future reasoning model releases potentially outperform R2TL? I still did appreciate the thorough evaluation, but as a user, would the author recommendation be to use R2TL right now, or wait for Deepseek-R1+N? That said, it seems like future R2TL can benefit from a vanilla DeepSeek reasoner and/or coder model (qwen2.5), so perhaps there is lots of merit to R2TL. Could the authors comment a bit about such a scenario?

---

> ### Author Rebuttal · Authors · 2025-07-30
>
> Thank you sincerely for your thoughtful and detailed feedback on our work. We greatly appreciate your recognition of the importance of the RTL code generation problem, as well as your positive remarks on our adoption of various ML-based techniques. Your inquiries regarding model applications have offered valuable guidance for both the current implementation and future development of R2TL. Addressing your primary concern, we present our perspectives below.
>
> - **Q1:** The results aren't mindblowing compared to state-of-the-art methods. So the question then becomes is such a complex approach necessary, or will future reasoning model releases potentially outperform R2TL? I still did appreciate the thorough evaluation, but as a user, would the author recommendation be to use R2TL right now, or wait for Deepseek-R1+N? That said, it seems like future R2TL can benefit from a vanilla DeepSeek reasoner and/or coder model (qwen2.5), so perhaps there is lots of merit to R2TL. Could the authors comment a bit about such a scenario?
>
> Thank you for raising this point. The core value of R2TL lies in its unique advantages that make it a compelling choice for specific use cases.​
>
> Firstly, R2TL boasts significant deployment advantages due to its compact size. With only 7B parameters, it stands in stark contrast to DeepSeek-R1’s 671B parameters, reducing storage requirements by two orders of magnitude. Additionally, its computational demands are approximately an order of magnitude lower than DeepSeek-R1 under the same token budget. Importantly, R2TL is already well-equipped to handle a range of automated design tasks effectively. These characteristics—strong capability paired with high deployment efficiency—make it a viable and excellent choice for local deployment, as it can be set up and run with minimal resource investment, allowing for quick integration into existing workflows.​
>
> Secondly, R2TL represents a generalizable approach that can evolve alongside advancements in general-purpose models, showing the effectiveness of the distillation followed by RL approach in the hardware design domain. Its performance is inherently tied to the capabilities of the models in its training pipeline—both distillation models like DeepSeek-R1 and base models like Qwen2.5-Coder-Instruct. As these models improve, with distillation models generating higher-quality training data and base models providing stronger foundational architectures, R2TL’s performance will strengthen in tandem. This adaptability allows it to leverage future developments in such models, ensuring it remains relevant and powerful as the field progresses.​
>
> Thirdly, in terms of user applicability, R2TL is particularly well-suited for professional developers and researchers with specialized needs. While entry-level users and developers may find general-purpose models more accessible via web interfaces or APIs to meet basic requirements, R2TL offers unique value to professionals. Specifically, when these users possess private data or their own base models, they can leverage R2TL's training pipeline to develop dedicated models that are more powerful and better suited to their specific needs. This makes it highly beneficial for those in fields like chip automated design or other AI domains to tackle domain-specific design tasks effectively.
>
> In summary, R2TL is not only a practical tool for current applications with its deployment-friendly size and efficiency but also a generalizable approach that can grow with advancements in foundational models, making it particularly valuable for professional users requiring independent deployment or training with their unique data.
>
> Thank you again for this valuable discussion. We welcome further discussion if you have any questions.

---

### Note · Authors · 2025-08-12

Dear ACs and reviewers,

Thank you so much for your time and valuable feedback during the rebuttal discussion. We have made the following contributions:

**Our rebuttal**:

In our rebuttal, we have made earnest and significant efforts, including conducting additional experiments and making further clarification. Reviewer ukvi has no further questions after our rebuttal. Reviewer NGaP has discussed with us and raised the score for addressing his/her concern. Reviewer oSjb has also raised the score for addressing most concerns. For his/her remaining concern, the relevance of Definition 2.1 and Theorem 2.1 to the rest of the paper, we have further explained the space definition and the intuition behind the theorem. We suggest this explanation can further help address reviewer oSjb's concern.

We conclude the rebuttal details as follows:

We have conducted detailed data analysis and case study on our SFT and RL datasets, VerilogEval and RTLLM benchmarks, and our rule-based test bench generation framework. We have conducted new experiments for agentic methods (MAGE). We have added detailed explanations on the practicability of our model and method, the sufficient quantity of our RL data, the generalization ability of our method, the meaning of our Theorem 2.1 and its connection to our method, and the ability of our natural language description generator.

**Our contributions**:

Our paper has the following contributions agreed by the reviewers:

(1) Our work addresses an important problem, RTL generation, with RLVR (ukvi, NGaP).

(2) Our work utilizes proper methods (as said by reviewer ukvi, “I particularly enjoyed the plethora of different ML-based techniques for this system's level problem.”), including round-trip data synthesis (ukvi, NGaP, oSjb), adaptive DAPO (ukvi, NGaP), testbench (NGaP).

(3) Our work achieves good results on benchmarks (ukvi, NGaP) with a cost-effective approach (oSjb).

(4) Our rebuttal addresses the concern about the different performance between VerilogEval and RTLLM (NGaP, oSjb), the scalability of our rule-based testbench framework (NGaP), the train dataset complexity (NGaP, oSjb), integration with agent methods (NGaP), and model and method applicability (ukvi).

Thank you for your time again!

Sincerely,

Authors of submission "R$^2$TL: Reasoning-Enhanced RTL Generation"

---

### Decision · Program_Chairs · 2025-09-17

**Decision:**

Accept (poster)

**Comment:**

The paper deals with training language models to generate register-transfer level (RTL) code, an interesting and important application in electronic design automation. Broadly it centers on ways to adapt the reinforcement learning with verified rewards (RLVR) framework to this setting, which lacks a verifier and is data-scarce. The paper proposes an automated testbench generation method based on insight into the problem domain (e.g., using circuit-structure analysis and simulations), as well as a round-trip data synthesis method.

The reviewers were all positive, with two accepts and one borderline accept. The problem space is interesting, the method has promising results, and the techniques are interesting and a nice fit for the problem. A remaining concern with one reviewer was the relevance of Definition 2.1 and Theorem 2.1, which should be clarified in the final version. Otherwise the reviewers remain quite positive about this paper. I agree and think that it would be a nice addition to NeurIPS. I recommend acceptance.